

**Marine Carbohydrates in Arctic Aerosol Particles and Fog – Diversity**
**of Oceanic Sources and Atmospheric Transformations**
**Sebastian Zeppenfeld[1], Manuela van Pinxteren[1], Markus Hartmann[2], Moritz Zeising[3], Astrid**
**Bracher[3,4], and Hartmut Herrmann[1]**
1 Atmospheric Chemistry Department (ACD), Leibniz-Institute for Tropospheric Research (TROPOS),
Leipzig, Germany
2 Atmospheric Microphysics (AMP), Leibniz-Institute for Tropospheric Research (TROPOS), Leipzig,
Germany
3 Alfred-Wegener-Institute Helmholtz Centre for Polar and Marine Research, Bremerhaven, Germany
4 Institute of Environmental Physics, University of Bremen, Bremen, Germany
*Correspondence to*: Hartmut Herrmann (herrmann@tropos.de)

# 18 Abstract

We present the results of a ship-based field study about the sea-air transfer of marine combined
carbohydrates (CCHO) from concerted measurements of the bulk seawater, the sea surface microlayer
(SML), aerosol particles and fog. In seawater, CCHO ranged between 22–1070 $\mu g\ L^{-1}$ with large
differences among the different sea-ice related sea surface compartments: ice-free ocean, marginal
ice zone (MIZ), open leads/polynyas within the pack ice and melt ponds. Enrichment factors in the SML
relative to the bulk water were very variable in the dissolved ($EF_{SML,dCCHO}$: 0.4–16) and particulate
($EF_{SML,pCCHO}$: 0.4–49) phases with highest values in the MIZ and aged melt ponds. In the atmosphere,
CCHO appeared in super- and submicron aerosol particles ($CCHO_{aer,super}$: 0.07–2.1 ng $m^{-3}$;
$CCHO_{aer,sub}$: 0.26–4.4 ng $m^{-3}$) and fog water ($CCHO_{fog,liquid}$: 18–22000 $\mu g\ L^{-1}$;
$CCHO_{fog,\ atmos}$: 3–4300 ng $m^{-3}$). The enrichment factors for the sea-air transfer were calculated for
super- and submicron aerosol particles and fog, however strongly varied depending on which of the
sea-ice related sea surface compartments was assumed as the oceanic emission source. Finally, we
observed a quick atmospheric aging of CCHO after their emission with indications for both
biological/enzymatic processes (based on very selective changes within the monosaccharide
compositions of CCHO) and abiotic degradation (based on the depolymerization of long-chained CCHO
to short free monosaccharides). All in all, the present study highlights the diversity of marine emission
sources in the Arctic Ocean and atmospheric processes influencing the chemical composition of
aerosol particles and fog.



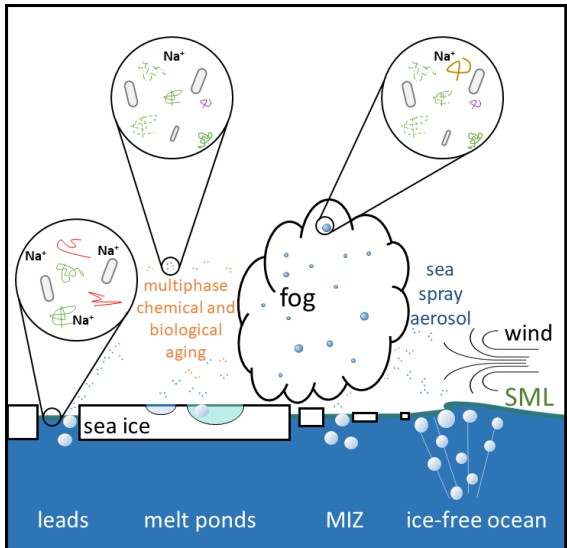

**TOC Figure**



## 1. Introduction

Sea spray aerosol (SSA) is one of the major aerosol species in the lower troposphere over the remote Arctic Ocean, particularly during the boreal spring and summer months (Chi et al., 2015; Hara et al., 2003). Depending on the size distribution and chemical composition, SSA strongly contributes to the populations of cloud condensation nuclei (CCN) and ice nucleating particles (INP) affecting the polar radiative budget through the formation of liquid droplets and ice crystals in fog and clouds (DeMott et al., 2016; Lawler et al., 2021; McCluskey et al., 2018; Penner et al., 2001; Schiffer et al., 2018; Wilbourn et al., 2020). Notably in the Arctic, one of the regions most affected by global warming, there is still a lack of knowledge about the relationship between the formation and evolution of clouds and specific chemical properties of primary marine aerosol particles (Wendisch et al., 2023).

SSA is emitted directly from the ocean surface through wind-driven processes and, as a consequence, contains the salts and the organic matter (OM) present in seawater, including carbohydrates (CHO) as one of the largest identified organic fractions (Quinn et al., 2015 and references therein). In microalgae, bacteria and also more complex marine organisms (e.g. kelp, krill), carbohydrates have important metabolic, structural and protective functions or are released in response to environmental stress, such as freezing or lack of nutrients (Krembs et al., 2002; Krembs and Deming, 2008; McCarthy et al., 1996; Mühlenbruch et al., 2018; Suzuki and Suzuki, 2013; Wietz et al., 2015). In seawater, the majority of carbohydrates appears as linear or branched oligo- and polysaccharides, commonly referred to as combined carbohydrates (CCHO), both in the dissolved ($d$CCHO) and the particulate ($p$CCHO) phases. These macromolecules consist of several monosaccharides, such as hexoses, pentoses, deoxy sugars, amino sugars, uronic acids and amino sugar acids, which are connected via glycosidic bonds (Benner and Kaiser, 2003; Engel and Händel, 2011; Panagiotopoulos and Sempéré, 2005). Most CCHO are quite stable within the marine environment unless they are either hydrolyzed in the presence of specific enzymes or in a very acidic setting (Arnosti, 2000; Panagiotopoulos and Sempéré, 2005). Heterotrophic bacteria use extracellular enzymes to selectively degrade CCHO into absorbable shorter molecules leaving a certain part as recalcitrant, more persistent OM (Alderkamp et al., 2007; Becker et al., 2020; Goldberg et al., 2011; Wietz et al., 2015). While $p$CCHO is mostly attributed to recent productions by local phytoplankton indicated by high positive correlations with total chlorophyll $a$ (TChl-$a$), $d$CCHO appears to be the result of more complex metabolic and transformation processes after its release (Becker et al., 2020; Fabiano et al., 1993; Goldberg et al., 2011; Zeppenfeld et al., 2021a). In contrast, dissolved free carbohydrates ($d$FCHO), short sugars in their monomer form, are quickly consumed by marine microorganisms resulting in much lower concentrations of $d$FCHO compared to CCHO in ambient seawater (Engbrodt, 2001; Engel and Händel, 2011; Ittekkot et al., 1981; Zeppenfeld et al., 2020).



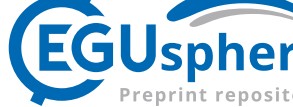

In the remote marine atmosphere, carbohydrates are assumed to significantly impact cloud properties
by contributing to both the CCN and INP populations (Leck et al., 2013; Orellana et al., 2011; van
Pinxteren et al., 2022). Carbohydrates appear both in super- and submicron SSA particles (Aller et al.,
2017; Leck et al., 2013; Russell et al., 2010; Zeppenfeld et al., 2021a), most likely resulting from their
emission from the surface of the ocean after bubble bursting as part of jet and film droplets (Veron,
2015; Wang et al., 2017). In addition to the bulk surface seawater, the sea surface microlayer (SML) as
the uppermost layer of the oceanic water column is an important source of OM, and thus marine
carbohydrates, in the SSA. The SML is described as a gelatinous film on top of the ocean, which is often
enriched in surface-active substances or buoyant gel particles compared to the underlying bulk water
(Engel et al., 2017; Wurl et al., 2009, 2011; Zäncker et al., 2017). Entrained air bubbles rise within the
upper part of the water column, scavenges surface-active organics from the surface bulk seawater and
pass the thin SML. Eventually they burst there releasing film and jet droplets containing a mixture of
substances found within the bulk and the SML (Burrows et al., 2014). At the same time, surfactants,
exopolymers and microgels in the SML increase the stability of the cap films of the bubbles, extend
their lifetimes and enable the drainage of water-soluble compounds (Bigg and Leck, 2008; Bikerman,
2013; Sellegri et al., 2006). Consequently, the sea-air transfer occurs in a chemo-selective manner
leading to a strong size-dependent enrichment of surface-active organics relative to water-soluble
sodium ($Na^+$) and, hence, a relative chemical composition of SSA different to the surface seawater
(Facchini et al., 2008; O'Dowd et al., 2004; van Pinxteren et al., 2017; Prather et al., 2013; Quinn et al.,
2015; Triesch et al., 2021a, b). These chemo-selective enrichments of organic substances in the SSA
relative to bulk water, especially in the submicron size range, usually exceed the enrichments in the
SML by orders of magnitude (van Pinxteren et al., 2017; Schmitt-Kopplin et al., 2012). The underlying
mechanisms for the chemo-selective sea-air transfer of carbohydrates, including co-adsorption, are
complex and subject of several recent and ongoing laboratory tank and modelling studies (Burrows et
al., 2016; Hasenecz et al., 2020; Schill et al., 2018; Xu et al., 2023). After their emission, fresh SSA
particles, including the contained carbohydrates, undergo atmospheric aging due to a not yet well-
understood interplay of several atmospheric processes, such as atmospheric acidification, abiotic
radical chemistry and biological and enzymatic modifications (Angle et al., 2021; Hasenecz et al., 2020;
Malfatti et al., 2019; Trueblood et al., 2019; Zeppenfeld et al., 2021a), potentially also altering their
microphysical properties.
Besides SSA, high concentrations of marine carbohydrates in fog and low-level clouds in the marine
environment are plausible due to the high hygroscopicity of SSA serving as good CCN (Xu et al., 2022)
transferring OM from the particle into the liquid phase, the high water-solubility of carbohydrates and
cloud-borne microorganisms potentially forming carbohydrates in-situ (Matulová et al., 2014). Only a
few studies conducted at field sites exposed to marine air masses measured certain subgroups of



carbohydrates, such as primary saccharides (Dominutti et al., 2022) or transparent exopolymer
particles (TEP) (Orellana et al., 2011; van Pinxteren et al., 2022) so far in fog/clouds. However, the
sources of marine carbohydrates in marine ambient fog/clouds, including $d$FCHO$_{fog}$ and CCHO$_{fog}$, and
their relationship to the bulk seawater, SML and aerosol particles still lack elucidation.
During the summer months, the chemical compounds of natural SSA and marine fog can be studied in
the Arctic Ocean due to the low influence of long-range transported anthropogenic pollution (Bozem
et al., 2019; Schmale et al., 2021). However, the presence and seasonal evolution of Arctic sea ice
divides this pristine region into a complex ensemble of several sea-ice related sea surface
compartments. These encompass the open leads - sea ice fractures with variable widths of several
meters - and polynyas within the pack ice. Furthermore, there is the ice-free ocean, the marginal ice
zone (MIZ) defined via a sea ice concentration threshold between 15 and 80% (Rolph et al., 2020), and
melt ponds forming and developing during the melting season on top of the ice floes. These
environments are characterized by different chemical, physical and biological characteristics
potentially influencing the quantity and properties of the SSA emitted. Recent studies observed, for
instance, that the number and efficiency of Arctic INP are strongly dominated by the type of sea-ice
related sea surface compartments that the air masses had passed before sampling (Creamean et al.,
2022; Hartmann et al., 2021; Papakonstantinou-Presvelou et al., 2022; Porter et al., 2022). However,
the individual conclusions still appear controversial and might be biased by seasonal and interannual
variabilities. Consequently, more systematic studies in the Arctic, also with regard to the chemical
properties of the aerosol particles, are required to achieve more conclusive results.
To increase the knowledge about marine carbohydrates as important constituents of SSA and potential
CCN and INP, we present here the results of a comprehensive field study conducted onboard the
German icebreaker RV *Polarstern* from May to July 2017. We performed concerted measurements of
bulk seawater, SML, size-resolved aerosol particles and fog water at different locations dominated by
different sea-ice related sea surface compartments (ice-free, leads/polynyas, MIZ, melt ponds) in the
Arctic Ocean. All marine and atmospheric compartments are discussed and compared on absolute
CCHO concentrations, calculated CCHO/Na$^{+}$ ratios, the relative monosaccharide contribution to CCHO
and the occurrence of $d$FCHO. Eventually, we disclose the complexity of the primary emission
mechanisms and subsequent atmospheric aging of marine CCHO in the Arctic Ocean.



## 2. Experimental

### 2.1 Study area and field sampling

Field samples were gathered during the PS106 (PASCAL/SiPCA) campaign (Macke and Flores, 2018; Wendisch et al., 2018) conducted from May to July 2017 in the Fram Strait, Barents Sea and central Arctic Ocean including an ice floe camp period (05–14 June 2017) on board the German icebreaker RV *Polarstern*.

Marine SML and corresponding bulk water samples were collected at various locations (**Figure SI 1**) from the ice-free ocean (four sampling events), open leads and polynyas within the pack ice (20 sampling events), young and aged melt ponds (six sampling events) and the MIZ (five sampling events). SML samples were obtained by immersing a glass plate (length: 50 cm, width: 20 cm, thickness: 0.5 cm, sampling area: 2000 cm$^2$) vertically into the surface water and slowly withdrawing it at a speed of approximately 15 cm s$^{-1}$ (van Pinxteren et al., 2012; Zeppenfeld et al., 2021a). The adhered SML film was drawn off the glass plate surface into a prewashed wide-neck plastic bottle by a framed Teflon wiper. The average thickness of the SML collected during this field study was 76±10 µm. The corresponding bulk water was taken from a defined depth of 1 m into LDPE bottles attached to a telescopic rod, except at the closed melt ponds where it was scooped from the bottom at approximately 20–40 cm depth. Before each sampling, the sampling containers were first rinsed with a few milliliters of the corresponding aqueous sample which was disposed immediately after. On board, small aliquots of the water samples were analyzed immediately for salinity using a conductivity meter (pH/Cond 3320, WTW), colored dissolved organic matter (CDOM) and particulate absorption (PAB), with more details in section 2.6. For later chemical analyses (inorganic ions, pH, carbohydrates) 500–1000 ml of 0.2 µm filtered water sample, 0.2 µm polycarbonate filters and field blanks were stored at -20°C.

The sampling of ambient aerosol particles was conducted at the starboard side of RV *Polarstern* at the top of the observation deck at a height of approx. 25 m above sea level as already described in Kecorius et al. (2019). Size-segregated aerosol particles were sampled in five size ranges (stage 1: 0.05–0.14 µm, stage 2: 0.14–0.42 µm, stage 3: 0.42–1.2 µm, stage 4: 1.2–3.5 µm, stage 5: 3.5–10 µm aerodynamic particle diameter with a 50% cut-off) on aluminum foils by using two synchronized low-pressure Berner impactors (Hauke, Austria) with a flow rate of 75 L min$^{-1}$ and a sampling time of three to six days. To avoid the condensation of atmospheric water and subsequent microbial activities on the aluminum foils, a 3 m long heated tubes between the isokinetic inlets and the impactors reduced the relative humidity of the sampled air to 75-80%, when the ambient relative humidity was higher. During this field study, the difference of the temperatures of the ambient air at the inlet and the sampled air after





the heating never exceeded 9 K. Consequently, losses of semi-volatile compounds or changes by heat-
induced chemical reactions are expected to be neglectable. Furthermore, the Berner impactors were
thermally insulated by a polystyrene shell. After sampling, the foils were stored in aluminum containers
at -20°C until analysis. In this study, the results from stages 1-3, 4-5 and 1-5 were summed up as
submicron (sub), supermicron (super) and $PM_{10}$, respectively. Details about the size-resolved aerosol
particle samples and corresponding meteorological information are given in (**Table SI 1**, in total 15
complete sets of Berner foils).
Close to the aerosol sampling, fog was collected using the Caltech Active Strand Cloud Collector Version
2 (CASCC2) as described by Demoz et al. (1996). Bulk fog droplets were impacted on Teflon strands
with a diameter of 508 μm and collected into a prewashed Nalgene polyethylene bottle. The flow rate
was 5.3 $m^3$ min and the 50% lower cut-off was determined to be approximately 3.5 μm. Further
information about the 22 fog samples collected during the PS106 campaign including meteorological
information can be found in **Table SI 2** and in Hartmann et al. (2021).

## 182    2.2 Total aerosol particle mass concentrations

Before and after sampling, the aluminum foils were equilibrated (three days, 20°C, 50% relative
humidity) and weighed using a precise microbalance (Mettler Toledo XP2U, weighing error: ±4.6 μg).
Total particle mass concentrations ($mass_{aer, stage\ y}$) were calculated for each Berner stage as the ratio
between the difference of the absolute foil masses after and before sampling and the sampled air
volume. Afterwards, aluminum foils were divided for further chemical analyses.

## 188    2.3 OC/EC in aerosol samples

Organic carbon ($OC_{aer}$) and elemental carbon on Berner aerosol foils were determined as described by
Müller et al. (2010) using a two-step thermographic method (C/S MAX, Seifert Laborgeräte, Germany)
with a nondispersive infrared sensor.

## 192    2.4 Carbohydrates in aerosol particles, fog, seawater and melt ponds

Marine carbohydrates in the particulate (*p*CCHO, >0.2 μm) and dissolved (*d*CCHO/*d*FCHO, <0.2 μm)
including truly dissolved molecules and small colloids were quantified from seawater and melt pond
samples following the protocol presented by Zeppenfeld et al. (2020, 2021a) using high-performance
anion-exchange chromatography with pulsed amperometric detection (HPAEC-PAD) equipped with a
Dionex CarboPac PA20 analytical column (3 mm × 150 mm) and a Dionex CarboPac PA20 guard column
(3 mm × 30 mm). The monosaccharides fucose (Fuc), rhamnose (Rha), arabinose (Ara), galactose (Gal),
glucose (Glc), xylose (Xyl), mannose (Man), fructose (Fru), galactosamine (GalN), glucosamine (GlcN),
muramic acid (MurAc), galacturonic acid (GalAc), and glucuronic acid (GlcAc) were identified by their



retention times. *d*FCHO represent the sum of identifiable monosaccharides before, and *d*CCHO and
*p*CCHO additionally released after an acid hydrolysis (0.8 M HCl, 100°C, 20 h). CCHO is the sum of
*d*CCHO and *p*CCHO. CHO represents the sum of CCHO and *d*FCHO, and consequently encompasses all
carbohydrates measured within this study. **Figure 1** gives an overview of the here used carbohydrate-
related abbreviations. Marine carbohydrates in fog water and extracts from size-resolved aerosol
particles were measured with (CCHO$_{fog}$, CCHO$_{aer}$) or without (*d*FCHO$_{fog}$, *d*FCHO$_{aer}$) prior acid hydrolysis.

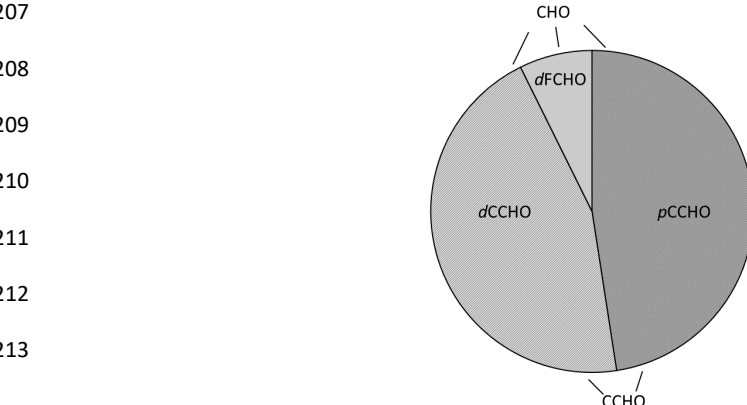

**Figure 1.** Overview of the abbreviations for carbohydrates (CHO) in seawater. CCHO: combined carbohydrates; *p*CCHO: particulate combined carbohydrates, *d*CCHO: dissolved combined carbohydrates; *d*FCHO: dissolved free carbohydrates.

## 2.5 Sodium and pH in aerosol particles, fog, seawater and melt ponds

Major inorganic ions, including sodium (Na$^+$), were determined from 0.45 µm filtered aqueous extracts
of the size-resolved aerosol samples (50% of the Berner foil in 2 ml ultrapure water), fog water, diluted
(1:15 000) seawater and melt pond samples using ion chromatography (ICS3000, Dionex) as described
by Müller et al. (2010). In this study, we discuss the results for Na$^+$ as a proxy for primary sea spray
emissions in remote marine regions. Additionally, the pH was monitored by an additional autosampler
sample conductivity and pH accessory (Dionex) in all seawater, melt pond and, whenever enough
sample volume was available, in fog water.

## 2.6 Absorption by phytoplankton, non-algal particles and colored dissolved organic matter in seawater and melt pond samples

For the investigation of bio-optical parameters in seawater and melt pond samples, the particulate
fraction was collected by filtering the water samples (5–500 ml) onto glass-fiber filters (GF/F,
Whatman), while the dissolved fraction was filtered through 0.2 µm Spartan syringe filters (Whatman,
Germany) immediately after sampling. The GF/F filters were analyzed to determine the absorption
spectra (i.e. 320–844 nm, 2 nm resolution) using the quantitative filtration technique with an
integrative-cavity absorption meter setup (QFT-ICAM) as developed by Röttgers et al. (2016). We





followed the protocol by Liu et al. (2018) for the instrument used here and the determination of the
absorption coefficients by total particles ($a_p440$), phytoplankton ($a_{ph}440$) and non-algal particles
($a_{NAP}440$) at λ=440 nm.
The absorption for the dissolved fraction ($a_{CDOM}(λ)$) between 270 and 750 nm (1 nm resolution) were
measured as triplicates using a long path length liquid waveguide capillary cell (LWCC) system following
the procedure by Lefering et al. (2017) and including the correction for salinity effects by Röttgers et
al. (2014) as described for our instrumentation in Álvarez et al. (2022). The absorption coefficients in
the visible at 443 nm ($a_{CDOM}443$) and UV at 350 nm ($a_{CDOM}350$) bands were used as indicators of CDOM
magnitude.

## 2.7 Supporting observations

The German research vessel *Polarstern* performs continuous meteorological surface measurements
during times of ship operation. For this study, we used the data from the HMP155
thermometer/hygrometer probe (Vaisala), the ultrasonic anemometer (Thies Clima) and the FS11
visibility sensor (Vaisala) each installed at a height of 29 m, 39 m and 20 m above sea level,
respectively. The quality-controlled data made available by the operators on the public repository
PANGAEA (Schmithüsen, 2018, 2019) supported the interpretation of the results of this study.
The 120 h back-trajectories were computed for the sampling periods of the size-resolved aerosol
particles and fog water events using the NOAA HYSPLIT model (Stein et al., 2015). The trajectories were
calculated on an hourly basis using the GDAS1 meteorological fields (Global Data Assimilation System;
1° latitude/longitude; 3-hourly) and at arrival heights of 50, 250 and 1000 m. Sea ice concentration
data were retrieved from ERDDAP (Environmental Research Division's Data Access Program), a data
server maintained by NOAA (National Oceanic and Atmospheric Administration). The MIZ was defined
here as the oceanic region with a sea ice concentration between 15 and 80%. Data on melt pond
fractions were accessed from the sea ice remote sensing data achieve of the University of Bremen
(https://data.seaice.uni-bremen.de, Istomina (2020)).

## 2.8 Statistics, calculations and visualization

Statistical analyses, calculations and visualization were performed in OriginPro, Microsoft Excel and R
version 4.2.1 using the following packages: oce, ocedata, ncdf4, openair, ggplot2, reshape2, scales,
lubridate, cmocean, maps, mapdata, rgdal, raster, RColorBrewer, sp. Time-resolved back-trajectories
and sea ice maps were combined using R to compute and visualize the air mass history regarding the
sea-ice related sea surface compartments that has been passed. Box-whisker plots represent the
interquartile range (box), median (horizontal line within the box), average (open square) and the
minimum and maximum values of the datasets (whiskers). Measured mean values are given together



with the calculated standard deviations (±). Correlations between two measured variables were
expressed via the Pearson correlation coefficient $R$. The thresholds of significance were set for the p-
values 0.1, 0.05, 0.01 and 0.001.
Enrichment factors for CCHO in the SML ($EF_{SML}$) relative to the corresponding bulk sample in different
sea-ice related sea surface compartments (ice-free, leads/polynyas, MIZ, melt ponds) were calculated
based on **Formula 1** with $[x]_{SML}$ and $[x]_{bulk}$ representing the concentrations of either $p$CCHO or $d$CCHO.
For the calculation of enrichment factors of CCHO in aerosol particles on Berner stage y ($EF_{aer,stage\ y}$;
**Formula 2**) and fog water ($EF_{fog}$; **Formula 3**) relative to the bulk water samples, the ocean was assumed
as the most likely source of atmospheric $Na^+$. For the calculations of $EF_{aer}$ and $EF_{fog}$, $CCHO_{bulk}$ and $Na^+_{bulk}$
concentrations of all individual bulk samples attributed to a certain sea-ice related sea surface
compartment (ice-free, leads/polynyas, MIZ, melt ponds) were averaged over the whole campaign.

$$EF_{SML} = \frac{[x]_{SML}}{[x]_{bulk}} \tag{1}$$

$$EF_{aer,stage\ y} = \frac{[x]_{aer,stage\ y}/[Na^+]_{aer,stage\ y}}{[x]_{bulk}/[Na^+]_{bulk}} \tag{2}$$


$$EF_{fog} = \frac{[x]_{fog}/[Na^+]_{fog}}{[x]_{bulk}/[Na^+]_{bulk}} \tag{3}$$




## 3. Results and Discussion

The sources of primary marine aerosol particles, and hence atmospheric marine carbohydrates, in the Arctic are diverse and influenced by the prevailing sea ice conditions. Here, we present the concentrations and relative compositions of CCHO in the SML and bulk water from the ice-free ocean, open leads and polynyas within the pack ice, melt ponds and the MIZ. After this, the different sea-ice related sea surface compartments are linked with the atmospheric CCHO found in ambient size-resolved aerosol particles and fog water. Eventually, the influence of the air mass history, enrichments of CCHO towards $Na^+$ during the sea-air transfer and secondary atmospheric transformations processes altering atmospheric CCHO are discussed.

### 3.1 Sea ice influences the properties of the sea surface water

***Variable CCHO concentrations in the Arctic surface water.*** CCHO were found in the dissolved ($d$CCHO) and particulate ($p$CCHO) phases of the SML and bulk water samples collected from the ocean and the melt ponds during the PS106 campaign. Among all these aqueous samples regardless of the sampling environment and depth (SML versus bulk), $d$CCHO (13–640 µg L$^{-1}$; mean$_{d$CCHO$}$ = 82±110 µg L$^{-1}$; n=70) and $p$CCHO (4–810 µg L$^{-1}$; mean$_{p$CCHO$}$ = 84±160 µg L$^{-1}$; n=70) were very variable. The occurring minima, maxima and mean values of both, $d$CCHO and $p$CCHO, however, ranged within the same orders of magnitude. CCHO as the sum of $d$CCHO and $p$CCHO ranged between 22–1070 µg L$^{-1}$ (mean$_{CCHO}$ = 166±250 µg L$^{-1}$; n=70).

Large differences in the mean values and standard deviations of CCHO were observed among the four sea-ice related sea surface compartments in the Arctic (leads/polynyas within the pack ice, MIZ, ice-free ocean, melt ponds) as shown in **Figure 2a+b**. The highest mean values for $d$CCHO and $p$CCHO were observed in the SML of the MIZ (mean$_{d$CCHO, SML, MIZ$}$ = 190±160 µg L$^{-1}$; mean$_{p$CCHO, SML, MIZ$}$ = 370±310 µg L$^{-1}$; n=5) and melt ponds (mean$_{d$CCHO, SML, melt ponds$}$ = 190±240 µg L$^{-1}$; mean$_{p$CCHO, SML, melt ponds$}$ = 200±310 µg L$^{-1}$; n=6), while the SML of the lead/polynya (mean$_{d$CCHO, SML, lead/polynya$}$ = 70±75 µg L$^{-1}$; mean$_{p$CCHO, SML, lead/polynya$}$ = 70±120 µg L$^{-1}$; n=20) and ice-free open ocean (mean$_{d$CCHO, SML, ice-free$}$ = 73±12 µg L$^{-1}$; mean$_{p$CCHO, SML, ice-free$}$ = 36±5 µg L$^{-1}$; n=4) samples tended to contain much less CCHO. The lower concentrations of the Arctic ice-free open ocean and the lead/polynya samples were rather similar to the ice-free part of the Southern Ocean west of the Antarctic peninsula (mean$_{d$CCHO, SML, Southern Ocean$}$ = 48±63 µg L$^{-1}$; mean$_{p$CCHO, SML, Southern Ocean$}$ = 72±53 µg L$^{-1}$; n=18; Zeppenfeld et al., 2021) during the austral summer, the tropical Cape Verde (mean$_{d$CCHO, SML, Cape Verde$}$ = 85±30 µg L$^{-1}$; van Pinxteren et al. (2023)) and the Peruvian upwelling region (mean$_{d$CCHO, SML, Peru$}$ ≈ 92±32 µg L$^{-1}$; Zäncker et al. (2017)). Consequently, the Arctic MIZ and melt ponds, especially the aged ones with




advanced microbiological activities, stood out with elevated CCHO within the Arctic and also compared
to tropical and other polar regions.

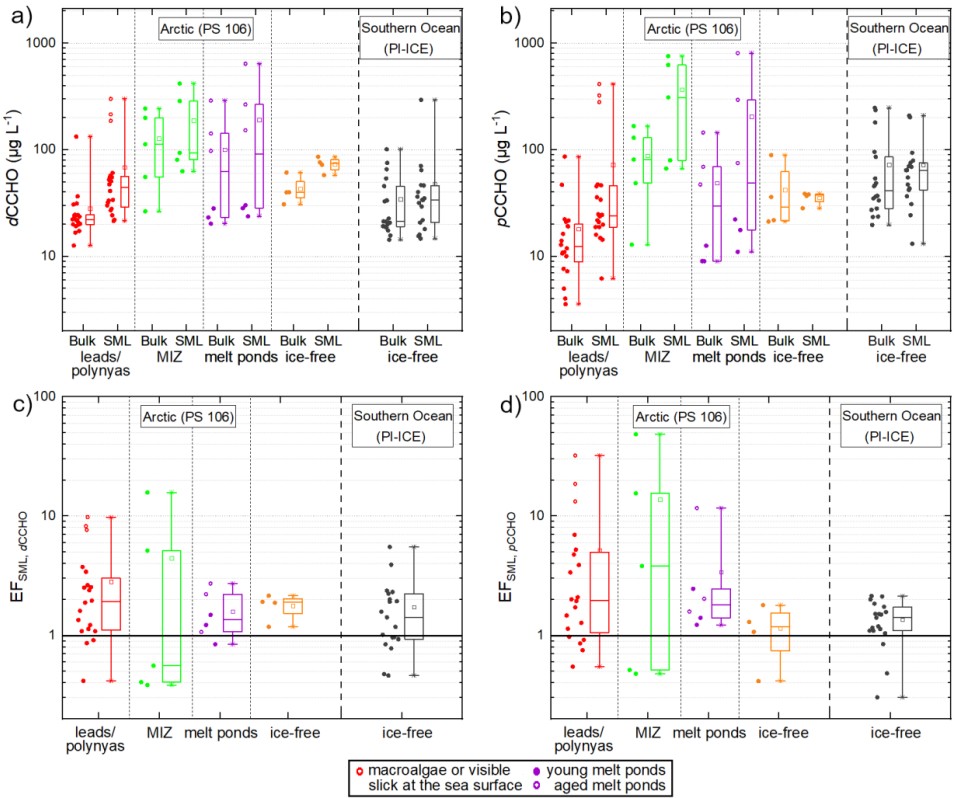

**Figure 2.** Scattered box-whisker plots showing the concentrations of a) $d$CCHO and b) $p$CCHO in the bulk and SML samples from the open leads and polynyas in the pack ice (red), the MIZ (green), ice-free open ocean (orange) and young and aged melt ponds (purple) collected during the PS106 campaign in the Arctic in comparison to the ice-free part of the Southern Ocean west of the Antarctic Peninsula investigated during the PI-ICE campaign in 2019 (black) as published in Zeppenfeld et al. (2021). EFs between SML and bulk water are shown in c) for $d$CCHO and d) for $p$CCHO. The black horizontal line represents an EF=1 meaning no enrichment or depletion.

***Variable enrichments of CCHO in the SML.*** The enrichment factors (EF$_{SML}$) of the CCHO in the SML
relative to the corresponding bulk water ranged between 0.4 and 16 for $d$CCHO (**Figure 2c**), while the
EF$_{SML}$ for $p$CCHO varied between 0.4 and 49 (**Figure 2d**). The vast majority, namely 80% of the SML
samples, was moderately until highly enriched in marine carbohydrates with only a few exceptions
where they were depleted (7 for $d$CCHO and 8 for $p$CCHO out of 35 in total). With a median
EF$_{SML,pCCHO,MIZ}$ of 3.8 and a mean of 13.8, the enrichment of $p$CCHO in the MIZ stood out in contrast to
the $p$CCHO of other sea-ice related sea surface compartments and $d$CCHO in general. However, it
should be noted that the number of MIZ samples was low and median and mean values were
dominated by three sample pairs with very high EF$_{SML}$ values. Low to moderate enrichments for $d$CCHO
and $p$CCHO were typically found in the lead/polynya samples from the pack ice (median



$EF_{SML,dCCHO,leads/polynyas}$=1.9; median $EF_{SML,pCCHO,leads/polynyas}$=2.0, n=20). However, three lead samples
showed quite high $d$CCHO & $p$CCHO concentrations in the SML compared to the corresponding bulk
samples resulting in high $EF_{SML,dCCHO,leads/polynyas}$ up to 10 and $EF_{SML,pCCHO,leads/polynyas}$ up to 32. The
exceptionally high EFs of these three samples can be explained by the observation of slicks - visible
films on the sea surface with altered reflectance and typically high enrichments of organics (Cunliffe
et al., 2013; Stolle et al., 2010; Williams et al., 1986; Wurl et al., 2009) as well as the presence of
macroalgae floating at the ocean's surface near the sampling site. Even though the macroalgae were
not collected themselves, their exudates or fragments might have been released, accumulated and
distributed in the SML close-by and thus sampled. Consequently, the few samples with high EFs in open
leads might rather represent exceptional events as spatially small-scale phenomena.
The slight to high enrichments for $d$CCHO and $p$CCHO in this study are in good agreement with the
values reported by Gao et al. (2012), who determined $EF_{SML,\,dCCHO}$ between 3.5 and 12, and $EF_{SML,\,pCCHO}$
between 1.7 and 7.0 for open leads within the central Arctic Ocean. Furthermore, the $EF_{SML,\,dCCHO}$ of the
four Arctic sea ice-related sea surface compartments reported here were not significantly different
compared to values found in the ice-free part of the Southern Ocean (ANOVA, one-way, 0.05
significance level). For the $p$CCHO, however, the average $EF_{SML}$ in the Arctic MIZ was significantly higher
than the one of the Southern Ocean, whereas the $EF_{SML,\,pCCHO}$ of the ice-free ocean in the Arctic were
similar to the Southern Ocean.
For explaining the accumulation in the SML, previous studies proposed several mechanisms and
processes, which fundamentally differ for the dissolved and particulate carbohydrates. The enrichment
of $p$CCHO in the SML might be dominated by an interplay of density-related and wind-driven processes.
For instance, the positive buoyancy of TEP, a subgroup of $p$CCHO, leads to an upward flux serving as a
continual vehicle for marine organisms and attached chemical compounds (Azetsu-Scott and Passow,
2004; Mari et al., 2017). Furthermore, strong winds can cause a short-term mixing of the upper water
column reducing the $EF_{SML}$ of particulates (Obernosterer et al., 2008) or TEP (Wurl et al., 2009; Zäncker
et al., 2021), while the wind-induced entrainment of air and the bubbling of seawater convert dissolved
negatively charged $d$CCHO and colloids into larger aggregates due to their sticky properties leading to
an enrichment of $p$CCHO in the SML (Passow, 2002; Robinson et al., 2019; Wurl et al., 2011). The
enrichment of $d$CCHO and also $d$FCHO in the SML is attributed to co-adsorption to other surface-active
compounds from the seawater matrix being scavenged at the surface of rising bubbles (Burrows et al.,
2016; Hasenecz et al., 2020; Schill et al., 2018; Xu et al., 2023). Additionally, microbial processes in the
SML could enhance the enrichment by in-situ formation and release of $d$CCHO by micro- or
macroalgae, while photolysis and enzymatic degradation of $d$CCHO to $d$FCHO by heterotrophic
bacteria would lead to a reduction of the enrichment in the SML. Specific to the Arctic, the release of



meltwater from the sea ice could be an additional source for carbohydrates in the SML, considering
the production of CCHO, exopolymeric substances (EPS) and TEP by sea ice algae and bacteria as a
protection strategy against freezing damage and fluctuating salinity in sea ice (Aslam et al., 2016;
Galgani et al., 2016; Krembs et al., 2002; Krembs and Deming, 2008). This could explain the
extraordinarily high $EF_{SML}$ observed in some, but not all, samples from the MIZ and melt ponds. In
summary, several processes might be responsible for enrichment processes in the SML, especially in
the Arctic, where the melting of sea ice could strongly bias the physiochemical processes usually
observed in controlled tank experiments.
***High and low salinities due to freezing and melting of sea ice.*** While the surface seawater of the Arctic
Ocean is very saline, the Arctic sea ice is much fresher due the separation into salt-free ice crystals and
a salty brine during its formation from seawater and a subsequent salt loss from gravity drainage in
winter and flushing during summer (Notz and Worster, 2009). During the late spring and summer
period of this study, when strong melting of sea ice occurs, a large amount of freshwater enters the
surface of the ocean creating inhomogeneities of salinity within the surface of the ocean. In both the
ice-free ocean and the pack ice, where sea ice exists, but the melting rate is low, salinities of the SML
and the bulk water ranged in this study between 30.9 and 34.5 (Zeppenfeld et al., 2019b), which is
typical for the SML and the surface bulk water of the Arctic Ocean (Vaqué et al., 2021). Within the MIZ,
where freshwater from melting sea ice quickly mixes with the salty ocean water, salinities were similar
with values in this study between 30.1 and 33.4, however, also with an exception in the SML of 25.7.
Melt ponds that were not yet joined at the bottom with the ocean below, were much fresher with
lower and more variable salinities ranging from 4.3 to 19.5 (Zeppenfeld et al., 2019b). With a few
exceptions, salinity discrepancies between the SML and the corresponding bulk water were small in
most cases.
Sea-air transfer studies usually refer to open ocean scenarios with high salinities in the seawater and
without the presence of melting sea ice. For the calculation of enrichment factors of organics in aerosol
particles ($EF_{aer}$) or fog ($EF_{fog}$), the concentrations of $Na^+$ – a major compound of sea salt – in the
seawater bulk is included by default (see equations 2 and 3). However, the Arctic is a more complex
marine environment where salinities, and hence $Na^+$ concentrations, can vary widely as melting
progresses. This may strongly influence the mechanisms behind the bubble bursting process, the
$CCHO/Na^+$ ratios in the bulk seawater and the SML, and thus also the $EF_{aer}$ and $EF_{fog}$ as it will be
discussed in section 3.4. Consequently, the variability of salinity in Arctic seawater and melt ponds
should be considered for sea-air transfer studies that rely on $Na^+$ values.
***Four sea-ice related sea surface compartments with different characteristics.*** In a nutshell, the high
Arctic differs from other oceanic regions in the presence, formation and melting of sea ice creating



sea-ice related sea surface compartments (ice-free, leads/polynyas, MIZ, melt ponds) with individual
biological and chemical characteristics, such as CCHO concentrations, enrichments in the SML and
salinities. This might potentially impact the transfer of substances from the ocean to the atmosphere,
chemo-selective enrichment processes of marine CCHO in the primary marine aerosol particles and
thus their microphysical properties. The next chapters will elucidate if and how these differences
within the individual compartments relate to $CCHO_{aer}$ and $CCHO_{fog}$.



### 3.2 Sea spray aerosol and therein contained combined carbohydrates

***Breaking waves as the main mechanism for SSA emissions is not unambiguous in the Arctic.*** In the
open ocean, the emission flux of SSA and hence its inorganic and organic constituents mainly depends
on the wind speed as the driving force for breaking waves and bubble bursting, and furthermore on
the seawater temperature, salinity, wave properties and organic surface-active substances (Grythe et
al., 2014). In this study, atmospheric sodium ($Na^+_{aer,PM10}$), the best tracer for SSA (Barthel et al., 2019),
ranged between 12 and 765 ng m$^{-3}$ (**Table SI 1**). $Na^+_{aer,PM10}$ showed a good correlation (R=0.80, p<0.001,
**Figure SI 2a**) with wind speed, measured at the sampling site and averaged over the sampling time, if
all aerosol samples are included. However, the strength of this correlation decreased sharply (R=0.59,
p<0.1), when only samples collected over the MIZ and the pack ice were included, while the few
samples from the open ocean characterized by high $Na^+$ values were excluded. This is due to the
presence of sea ice in the high Arctic, which likely alters and conceals the classical wind-driven
mechanisms of breaking waves and bubble bursting resulting in SSA emission. Firstly, sea ice covers a
significant part of the Arctic Ocean strongly reducing the area releasing SSA. Secondly, the presence of
sea ice causes an attenuation of the high-frequency wind-sea waves, while longer waves, such as
swells, can remain (Thomson, 2022). Consequently, the effect of wind on the SSA emission mechanisms
within the open leads and the MIZ might be different than in the ice-free ocean. For those sea-ice
dominated compartments, alternative wind-independent sources of ascending bubbles were
suggested, such as melting sea ice nearby, respiration of phytoplankton or sea-air heat exchange below
the sea surface (Chen et al., 2022 and references therein). Thirdly, in contrast to other marine regions
with quite homogeneous ocean salinities, and hence sodium concentrations, the salinities among the
different Arctic sea-ice related sea surface compartments are more variable due to the melting of sea
ice. Previously, the results of a sea-air transfer tank experiment with artificial seawater showed the
influence of salinity on the relative particle number concentrations of emitted SSA for salinities below
15 – values especially relevant for melt ponds in the Arctic – while changes at higher salinities did not
result in a measurable effect (Zábori et al., 2012). Additionally, organics with potential surface-active
properties are very variable in these disparate Arctic environments, as discussed for CCHO in chapter
3.1. Organic surfactants can alter the ocean surface's ability to form whitecaps and the lifetime of
bubbles (Bigg and Leck, 2008; Callaghan et al., 2012; Grythe et al., 2014) and therefore SSA properties.

Finally, blowing snow over the sea ice could serve as an additional, non-oceanic source of atmospheric
$Na^+_{aer}$ when a certain air-temperature-dependent wind speed threshold is exceeded (Chen et al., 2022;
Yang et al., 2008). Consequently, connections and correlations for the release of SSA particles in the
heterogeneous high Arctic are more difficult to explore than other marine environments without sea



ice. It can be assumed that this complex setting does not only influence the release of the inorganic
constituents from seawater, but its organic compounds, such as CCHO, too.
***CCHO_aer distributed in all size modes.*** During the PS106 campaign, the overall atmospheric
concentrations of $CCHO_{aer,PM10}$ ranged between 0.5 and 4.7 ng m$^{-3}$ (**Table SI 1**). Combined
carbohydrates were found on both supermicron ($CCHO_{aer,super}$=0.07–2.1 ng m$^{-3}$) and submicron
particles ($CCHO_{aer,sub}$=0.26–4.4 ng m$^{-3}$). Thus, these $CCHO_{aer}$ values ranged within the same orders of
magnitude as in the Arctic studies by Karl et al. (2019) and Leck et al. (2013) or the study conducted at
the western Antarctic peninsula by Zeppenfeld et al. (2021a). $CCHO_{aer}$ appeared in all of the five size
classes in variable concentrations (**Figure 3a**). Although the average concentrations were similar on all
stages, local maxima were observed on stages 2 (0.14–0.42 μm) and 5 (1.2–10 μm). A similar size
distribution of marine $CCHO_{aer}$ in these specific size ranges, but more pronounced, has been already
observed in the ice-free part of the Southern Ocean by Zeppenfeld et al. (2021a) explaining these
findings with a likely release of marine polysaccharides from the ocean as part of film and jet droplets.
Possibly, the aerosol size distribution of marine polysaccharides resulting from wind-driven bubble
bursting emissions are not as obvious in this Arctic study as it was in the ice-free Southern Ocean due
to the presence of Arctic sea ice suppressing and altering the local SSA emission mechanisms as
indicated in the previous section. The relative contribution of $CCHO_{aer}$ to $mass_{aer}$ varied between 0.01%
and 4% (**Figure 3b**), while the carbon contained within the combined carbohydrates ($C$-$CCHO_{aer}$)
contributed 0.06 to 4.9% to the $OC_{aer}$ in the size-resolved aerosol particles (**Figure 3c**). These
contributions agree well with the findings in marine aerosol particles from the Southern Ocean
(Zeppenfeld et al., 2021a).

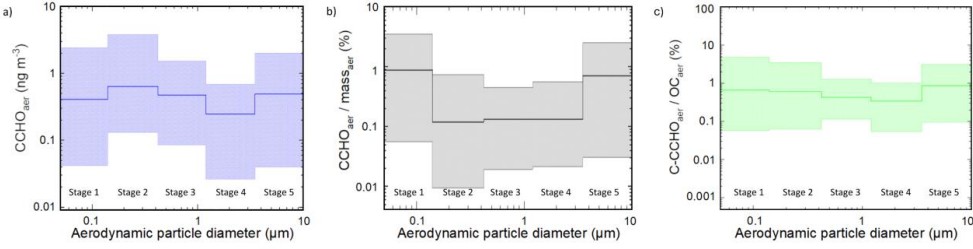

**Figure 3.** a) Concentration of combined carbohydrates in size-resolved aerosol particles ($CCHO_{aer}$), b) ratio of $CCHO_{aer}$ to the total particle mass concentration ($mass_{aer}$), c) ratios of carbon contained within the combined carbohydrates in aerosol particles ($C$-$CCHO_{aer}$) to organic carbon in aerosol particles ($OC_{aer}$). The bold lines represent the average concentrations during the PS106 campaign. The hatched areas show the range between the maximum and minimum values. The aerodynamic particle diameter refers to sampling conditions at relative humidity of max. 80%.

Unlike the study conducted in the Southern Ocean (Zeppenfeld et al., 2021a), $CCHO_{aer,PM10}$ in this study
showed no significant correlations with $Na^+_{aer,PM10}$ (R=0.24, p>0.1, **Figure SI 2b**) or wind speed (R=0.26,
p>0.1, **Figure SI 2c**), which could be due to the complex marine environment and the relevance of



several emission mechanisms in the Arctic as discussed above. However, if the correlations are
resolved for the different Berner impactor stages (i.e. size ranges), a large variability can be observed
(**Figure 4**). A higher correlation was found especially on stage 4 (1.2–3.5 µm) between $CCHO_{aer,stage\ 4}$
and $Na^{+}_{aer,stage\ 4}$ (R=0.76, p<0.01), while the Pearson correlations coefficients for the other Berner
stages were much lower. This could indicate the same marine source and wind-driven emission
mechanism for both chemical constituents in this supermicron aerosol size mode, while other aerosol
size modes might have been influenced by atmospheric aging and wind-independent emission
mechanisms as already mentioned for $Na^{+}_{aer}$ in the previous section. This observation agrees well with
the findings by Bigg and Leck (2008) and Leck (2002) reporting submicron polymer gel particles, likely
consisting of polysaccharides, in the atmosphere of the high Arctic containing almost no sea salt and
showing large similarities to those particles found in open leads close-by. This is quite surprising
considering that the mechanism of wind-driven wave breaking is quite limited due to the lack of long
fetches of open water (Held et al., 2011; Norris et al., 2011).

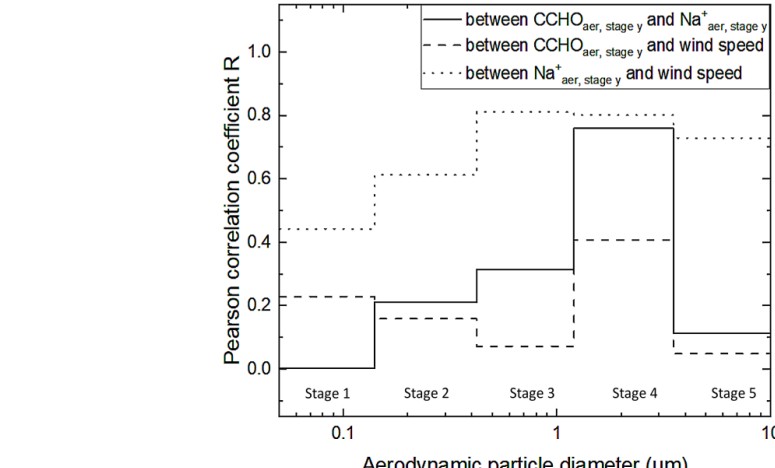

**Figure 4.** Pearson correlation coefficient R between $CCHO_{aer,stage\ y}$ and $Na^{+}_{aer,stage\ y}$ (solid line), between $CCHO_{aer,stage\ y}$ and the average wind speed (dashed line), and between $Na^{+}_{aer,stage\ y}$ and the average wind speed (dotted line) for each stage y of the Berner impactor.

Blowing snow has been discussed as a possible additional source for atmospheric $Na^{+}$, raising the
question, if it could be a source for atmospheric carbohydrates, too. During this study, the
measurements of $d$FCHO and CCHO in five Arctic snow samples collected resulting in low values mostly
below the limits of detection. This finding supports the conclusion that blowing snow does not serve
as a competitive source for the emission of atmospheric carbohydrates.



### 3.3 Marine combined carbohydrates in fog


The concentrations of $Na^+_{fog, liquid}$ (1.7–903 mg L$^{-1}$; mean = 130±220 mg L$^{-1}$; n=22) and $CCHO_{fog, liquid}$
(18–22000 µg L$^{-1}$; mean = 1380±4600 µg L$^{-1}$; n=22) were very variable in fog water (**Table SI 2**).
Atmospheric fog concentrations of these chemical constituents can be calculated under consideration
of the liquid water content (LWC) during the fog events. Since LWC was not measured during PS106
directly, the LWC was approximated from the measured CCN concentrations at the lowest quality
assured supersaturation of 0.15% and an assumed average droplet diameter of 17 µm resulting in a
LWC of 0.20±0.27 g m$^{-3}$ (Hartmann et al., 2021). Following this approach, atmospheric concentrations
in fog ranged between 0.12 and 150 µg m$^{-3}$ (mean = 25±43 µg m$^{-3}$; n=16) for $Na^+_{fog, atmos}$ and between
3 and 4300 ng m$^{-3}$ (mean = 390±1100 ng m$^{-3}$; n=16) for $CCHO_{fog, atmos}$, respectively. These atmospheric
concentrations in fog are for both $Na^+_{fog, atmos}$ and $CCHO_{fog, atmos}$ by one to three orders of magnitude
higher than the atmospheric concentrations in aerosols discussed in section 3.2. This divergence may
be explained by the following:
- Fog scavenging is a transfer process of aerosol particles into the liquid phase of fog droplets
(Gilardoni et al., 2014). As fog forms and grows, it can capture aerosol particles in the air and
increase their concentration within the fog droplets. This could lead to higher atmospheric
concentrations of aerosol particle compounds, especially for the water-soluble and
hygroscopic ones, inside the fog compared to the surrounding air.
- The activation of aerosol particles to fog droplets is a process dominated by particle size with
larger particles tending to activate first. It is conceivable that SSA particles larger than 10 µm,
usually few in number, but with a large mass contribution, were available near the sea surface,
where sampling occurred. These SSA particles were activated into fog droplets and contributed
significantly to the $Na^+$ and CCHO in the fog. In contrast, aerosol sampling was restricted by
the Berner impactor's 10 µm diameter cut-off neglecting the larger particles in the
consideration.
- The LWC values were not measured but estimated, which could be a source of errors.
However, the calculated LWC values ranged within a realistic frame for Arctic fog and likely are
not responsible for the large difference between aerosol and fog concentrations in several
orders of magnitude.
Since both, organic and inorganic constituents, showed higher atmospheric concentrations in
fog/clouds compared to ambient aerosol particles, we conclude that a physical phenomenon, such as
fog scavenging, might explain this observation and not an in-situ formation within the cloud droplets.
Similar to the findings of this study discussing marine CCHO and $Na^+$ in Arctic fog, Triesch et al. (2021a)



found strikingly high concentrations of free amino acids (FAA) and $Na^+$ in marine clouds compared to
aerosol particles both collected on top of the Mt. Verde on Cape Verde Islands as shown in **Table 1**.
While $d$FCHO$_{fog}$ and derivatives, such as anhydrosugars and sugar alcohols, have been readily reported
for fog water with terrestrial and marine background (Dominutti et al., 2022), we here present for the
first time ambient CCHO concentrations in marine fog.

**Table 1.** Atmospheric concentrations of selected SSA constituents in fog/clouds compared to ambient aerosol particles during marine field studies.

| Chemical constituent | Fog/cloud | PM$_{10}$ | Sampling location | Sampling height | Sampling period | Reference |
|---|---|---|---|---|---|---|
| | (ng m$^{-3}$) | (ng m$^{-3}$) | | | | |
| $d$FCHO | 9.2—52[a,b] | — | Reúnion | 1760 m a.s.l.[d] | March—April 2019 | Dominutti et al. (2022) |
| | 1.5—1040 (mean:80±260) | <LOD—2.0 | Arctic | 25 m a.s.l.[e] | May—July 2017 | **this study** |
| CCHO | 3—4300 (mean:390±1100) | 0.5—4.7 | Arctic | 25 m a.s.l.[e] | May—July 2017 | **this study** |
| FAA | 11—490 | 1.0—4.8 | Cape Verde | 744 m a.s.l.[f] | Sept.—Oct. 2017 | Triesch et al. (2021a) |
| | 6—79 | — | Reúnion | 1760 m a.s.l[d] | March-April 2019 | Dominutti et al. (2022) |
| | (µg m$^{-3}$) | (µg m$^{-3}$) | | | | |
| Na$^+$ | 1.6—7.2 | 0.17—0.40 | Cape Verde | 744 m a.s.l. [f] | Sept.—Oct. 2017 | Triesch et al. (2021a) |
| | 0.1—2.2[b] | — | Reúnion | 1760 m a.s.l.[d] | March—April 2019 | Dominutti et al. (2022) |
| | 0.014—0.063[c] | — | Arctic | 180-374 m[g] | Aug.—Sept. 2018 | Zinke et al. (2021) |
| | 0.12—150 (mean:25±43) | 0.012—0.77 | Arctic | 25 m a.s.l.[e] | May—July 2017 | **this study** |

[a]only includes free glucose and rhamnose; sugar alcohols and anhydrosugars were not included for this table. [b]values were
calculated from LWCs, molecular weights and concentrations in fog water given within the reference; terrestrial contributions
are likely. [c]calculated from concentration in fog water and an assumed LWC of 0.1 g m$^{-3}$. [d]Piste Omega. [e]RV *Polarstern*. [f]Mt.
Verde. [g]tethered balloon.




### 3.4 Chemo-selective sea-air transfer of marine carbohydrates

The chemo-selective sea-air transfer of organics towards inorganic sea salt constituents has been
described both in tank and ambient field studies for organic carbon in general (Gantt et al., 2011;
Hoffman and Duce, 1976; van Pinxteren et al., 2017) or several chemical constituents, such as
carbohydrates (Hasenecz et al., 2020; Schill et al., 2018; Zeppenfeld et al., 2021a), lipids (Triesch et al.,
2021b) and free and combined amino acids (Triesch et al., 2021a, c).The calculation of dimensionless
ratios between the concentrations of the examined organic parameter and $Na^+$ allows a comparison
of aquatic and atmospheric samples within the marine environment. **Figure 5** shows the CCHO/$Na^+$
ratios for the bulk and SML in the four sea-ice related sea surface compartments, size resolved aerosol
particles and fog water collected during the PS106 cruise.

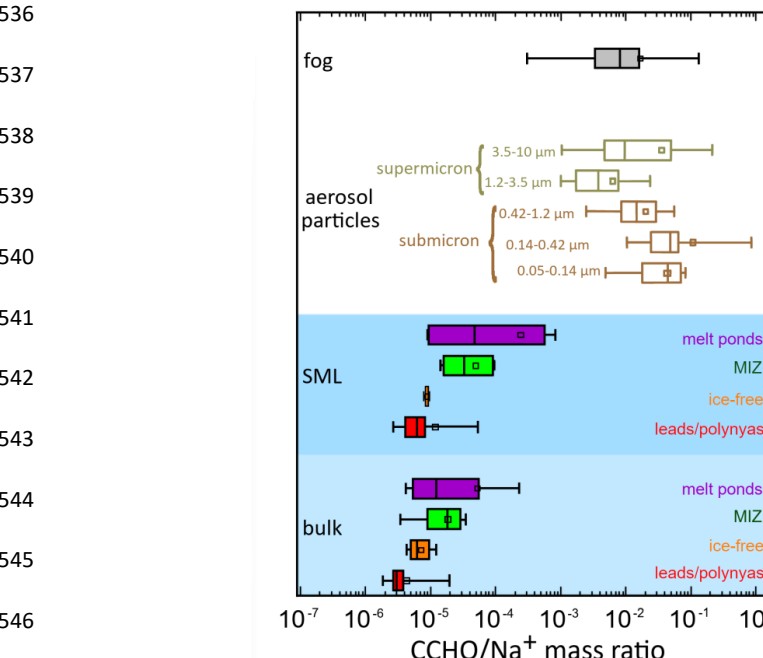

**Figure 5.** CCHO/$Na^+$ ratios for CCHO in Arctic fog, size-resolved aerosol particles and the surface seawater (SML and bulk)
from melt ponds, the marginal ice zone (MIZ), the ice-free ocean and leads/polynyas from the pack ice.

***Wide range of CCHO/$Na^+$ ratios in Arctic surface seawater.*** In the surface seawater samples of this
study, the CCHO/$Na^+$ ratios spanned from $2\times10^{-6}$ to $8\times10^{-4}$, representing a wider range than those
found in the Southern Ocean ($9\times10^{-7}$ and $3\times10^{-5}$; Zeppenfeld et al., 2021). While the ratios in the SML
and bulk water in general ranged in the same orders of magnitude, large differences were observed in
the individual Arctic sea-ice related sea surface compartments. In the SML, lowest median values were
found in the leads/polynyas and ice-free ocean samples with $6\times10^{-6}$ and $9\times10^{-6}$, respectively, while
higher median values appeared in the SML of the MIZ ($3\times10^{-5}$) and melt ponds ($4\times10^{-5}$), or even



$6×10^{-4}$, when only aged melt ponds were considered. This large variability of $CCHO/Na^+$ ratios can be
explained by the variable content of CCHO (high CCHO content in aged melt ponds & MIZ versus ice-
free ocean & leads/polynyas) and $Na^+$ (low salinity in the SML of melt ponds versus higher salinities in
ice-free ocean & leads/polynyas & MIZ) in the different sea-ice related sea surface compartments. It
can be expected, that the different $CCHO/Na^+$ ratios in the individual seawater compartments
impacted the corresponding $CCHO/Na^+$ ratios in fog and aerosol particles during the sea-air transfer,
and consequently the enrichment factors for the sea-air transfer ($EF_{aer}$, $EF_{fog}$), which are calculated
from those ratios.
***Air mass history influences $CCHO_{aer}/Na^+_{aer}$ ratios in Arctic aerosol particles.*** In contrast to the
seawater samples, $CCHO_{aer}/Na^+_{aer}$ ratios were much higher for aerosol particles considering the size
resolution ($1×10^{-3}$–$9×10^{-1}$) supporting the concept of the chemo-selective enrichment of
carbohydrates towards $Na^+$ during the transfer from the ocean into the atmosphere. In this context,
submicron particles showed much higher median ratios of $4×10^{-2}$ (0.05–0.14 µm) and $4×10^{-2}$
(0.14–0.42 µm) than supermicron particles with $4×10^{-3}$ (1.2–3.5 µm) and $1×10^{-2}$ (3.5–10 µm).
Considering $PM_{10}$ (sum of all five Berner stages), the $CCHO_{aer,PM10}/Na^+_{aer,PM10}$ ratios varied much more
in the Arctic study presented here ($2×10^{-3}$–$2×10^{-1}$, see **Table SI 1**) than in the ice-free part of the
Southern Ocean ($8×10^{-4}$–$7×10^{-3}$; Zeppenfeld et al. (2021b)). Interestingly, in the four aerosol sampling
periods (24/05/17–26/05/17; 26/05/17–29/05/17; 29/05/17–01/06/17; 19/06/17–25/06/17) where
the air masses had passed the majority (45–100%) of the previous 12 h before sampling over the open,
ice-free ocean (trajectories combined with sea ice maps can be found in **Figure SI 3**) exhibited the
lowest $CCHO_{aer,PM10}/Na^+_{aer,PM10}$ ratios ($2×10^{-3}$–$9×10^{-3}$, see **Table SI 1**), very similar to the values in the
ice-free Southern Ocean. In contrast, higher ratios were found, when the air masses had rested a
significant time over the pack ice or the MIZ. This could be an indication that the chemical composition
of the sea-ice related sea surface compartments, here the ice-free ocean with low $CCHO/Na^+$ ratios,
strongly influences the relative composition of aerosol particles. In contrast, the influence of the MIZ,
pack ice and melt ponds exhibiting quite different chemical, physical and biological properties on
$CCHO_{aer,PM10}/Na^+_{aer,PM10}$ could not be resolved in further details following this approach using trajectory
calculation and satellite data. This is certainly due to the proximity of these sea-ice related sea surface
compartments on a small spatial scale (especially melt ponds in direct vicinity to open leads), the long
sampling periods of aerosol particles, the lacking knowledge of deposition rates, the effect of wind on
bubble bursting processes within the individual sea-ice related sea surface compartments and missing
data on the biological activities in individual melt ponds.
***Similar $CCHO/Na^+$ ratios in aerosol particles and fog.*** For fog, $CCHO_{fog}/Na^+_{fog}$ ratios ranged from
$3×10^{-4}$ to $1×10^{-1}$, which covers the same orders of magnitude of aerosol particles. Even though absolute



atmospheric concentrations of CCHO are much higher in fog than in aerosol particles possibly due to
fog scavenging (as discussed in 3.3), the CCHO/Na$^+$ ratios are similar. This strongly implies that CCHO$_{fog}$
actually originated from the ambient marine aerosol particles. The attempt to find matches or common
trends between aerosol particles and the fog in individual samples was not successful, certainly due to
the very different resolutions of sampling times and in addition due to the probability of fog droplets
containing aerosol particles bigger than 10 μm.
***Calculated EF$_{aer}$ and EF$_{fog}$ depend on the sea-ice related marine source under consideration.*** EF$_{aer}$ and
EF$_{fog}$ are calculated as a quotient between the CCHO/Na$^+$ ratios in the size-resolved aerosol
particles/fog and the corresponding bulk water. The concentrations in the Arctic seawater of this study
was very variable depending on the regarded sea-ice related sea surface compartment environment,
as well in the aerosol particles and in fog water. This fact strongly impacted the resulting EF$_{aer}$ and EF$_{fog}$,
enabling calculated values ranging between $10^1$ and $10^4$ for supermicron aerosol particles, between
$10^2$ and $10^5$ for submicron particles and between $10^0$ and $10^4$ for fog depending on which sea-ice
related sea surface compartment was assumed as the marine source of SSA as shown in **Figure 6**.
Lower atmospheric EFs were calculated when aged melt ponds (EF$_{aer,super}$=19–750; EF$_{aer,sub}$=127–5100;
EF$_{fog}$=5–2400) or the MIZ (EF$_{aer,super}$=60–2310; EF$_{aer,sub}$=390–16000; EF$_{fog}$=17–7400) were assumed as
the only (theoretical) marine source of CCHO and Na$^+$, while higher values were found with the ice-
free ocean (EF$_{aer,super}$=175–6800; EF$_{aer,sub}$=1100–46000; EF$_{fog}$=50–22000) or open leads/polynyas
(EF$_{aer,super}$=360–14000; EF$_{aer,sub}$=2360–95000; EF$_{fog}$=103–44600). It is important to note that EFs were
most consistent with results from other CCHO sea-air transfer studies in the tank (Hasenecz et al.,
2020) and the field (Zeppenfeld et al., 2021a), when aged melt ponds or the MIZ were considered as
the oceanic emission source. If leads/polynyas and the ice-free ocean are regarded as the only emission
source, higher EF$_{aer}$ and EF$_{fog}$ values resulted, and hence the assumption of a stronger mechanistic
enrichment. It is highly unlikely whether an air mass package had been exclusively exposed to
leads/polynyas during its history, and not to aged melt ponds or the MIZ. Consequently, none of the
Arctic sea-ice related sea surface compartments discussed above should be neglected in the discussion
of sea-air transfer of organic substances.
During the same Arctic field campaign, Hartmann et al. (2021) investigated INP in ambient aerosol
particles and compared it to bulk and SML in seawater from all the different sea-ice related sea surface
compartments using similar EF$_{aer}$ calculations as reported here. They concluded that an enrichment of
3 to 5 orders of magnitude was necessary during the sea-air transfer to fully attribute atmospheric INP
to oceanic sources. Here, we show that such high EF$_{aer}$ and EF$_{fog}$ for organics, and hence marine
biogenic INP, can be calculated, e.g. when open leads/polynyas were referred to as the only oceanic
source. In summary, Artic air masses have been impacted by different types of sea-ice related sea





surface compartments before sampling, whereas it is still unclear which one has the biggest effect on
the chemical composition of the marine aerosol particles. This aspect should be considered when the
marine SSA constituents are modelled for the Arctic from remote sensing data.











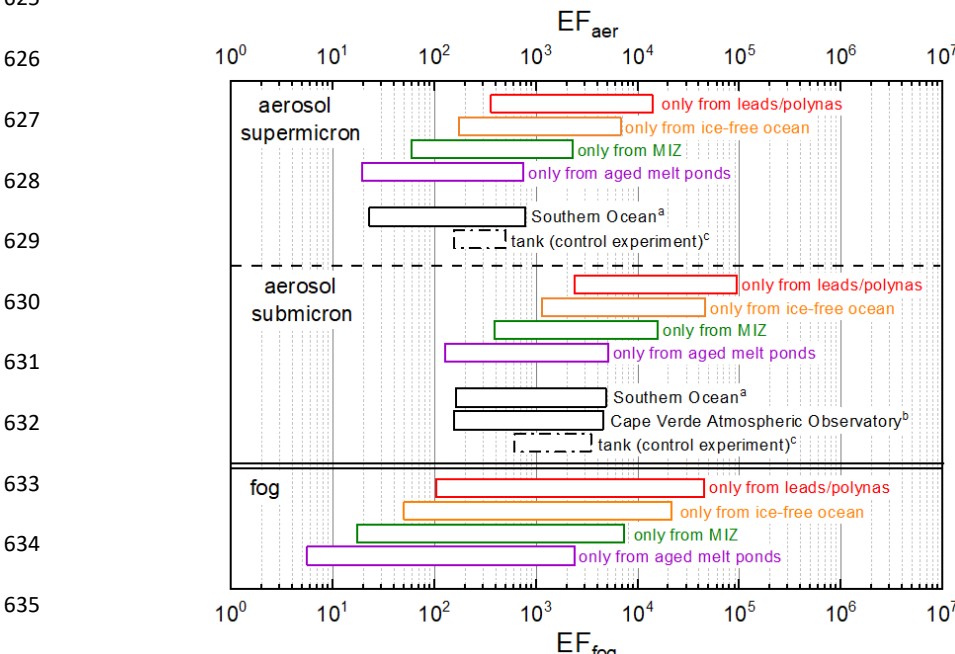

**Figure 6.** Range of calculated hypothetical enrichment factors EF$_{aer}$ and EF$_{fog}$ assuming either leads/polynyas, the ice-free ocean, the MIZ or aged melt ponds as the only marine source for the sea-air transfer of CCHO in the Arctic. For the calculation of EF$_{aer}$ and EF$_{fog}$, the minimum and maximum values of the CCHO$_{aer/fog}$/Na$^+_{aer/fog}$ ratios and the median values of CCHO$_{bulk}$/Na$^+_{bulk}$ were used. The EF$_{aer}$ values of this study were compared with the results of a) the field study conducted in the Southern Ocean by Zeppenfeld et al. (2021), b) the field study conducted at Cape Verde Atmospheric Observatory (CVAO) by van Pinxteren et al. (2023) and c) the results of the CCHO tank study by Hasenecz et al. (2020) without any addition of heterotrophic bacteria (control experiment). Here, EF$_{aer}$ values were calculated from the experimental data published by Hasenecz et al. (2020b).




### 3.5 Atmospheric aging of marine carbohydrates

To resolve the fate of marine carbohydrates in the atmosphere after their ejection from the ocean, the relative molar contribution of monosaccharides to CCHO were compared between the bulk and SML from the leads/polynyas, MIZ, ice-free ocean and melt pond samples, as well as the sub-and supermicron aerosol particles and fog water (**Figure 7**). The composition of marine carbohydrates in seawater strongly depends on the dominating microbial species, season, diagenetic state, availability of nutrients and environmental stress factors (Engbrodt, 2001; Goldberg et al., 2011) leading to a natural variability among individual samples even within small spatial scales. Consequently, to enable the direct comparison of seawater with atmospheric samples of this field study with an elevated level of statistical certainty, here we compare the mean values of the entire data set, instead of individual samples. Finally, in addition to the changes of the monosaccharide patterns of CCHO, the systematic degradation of CCHO to $d$FCHO was observed in the atmosphere and will be discussed within this chapter.

***CCHO composition in different sea-ice related sea surface compartments and depths is similar.*** In seawater (bulk and SML), glucose (means= 35–48 mol%), galactose (means= 13–18 mol%) and xylose (means= 7–16 mol%) dominated the CCHO composition followed by smaller contributions of other neutral sugars, amino sugars, uronic acids and muramic acid (**Figure 7**). Considering the natural variability among individual samples, there were no significant differences in means between the bulk and SML, nor between the lead/polynya, MIZ, ice-free ocean and melt pond samples. Variations were observed between the dissolved and particulate fractions (**Figure SI 6**), nevertheless the combined carbohydrates within all sea-ice related sea surface compartments followed the same pattern of the predominance of glucose, galactose and xylose. Overall, the relative monosaccharide compositions of glucose **>** (galactose ≈ xylose) **>** other (neutral or charged) monosaccharides of the seawater samples from this Arctic study appear similar to the monosaccharide compositions investigated in the SML and bulk water from the Central Arctic Ocean (Gao et al., 2012) and at the western Antarctic peninsula (Zeppenfeld et al., 2021a), the meltwater of Arctic multiyear sea ice (Amon et al., 2001) and the epipelagic water from the Ross Sea (Kirchman et al., 2001).

***Less galactose, but more muramic acid in atmospheric CCHO$_{aer}$ and CCHO$_{fog}$.*** Atmospheric samples showed a different monosaccharide pattern within the hydrolyzed CCHO in comparison to the seawater and melt pond samples. While glucose (means= 41; 50 and 60 mol% for fog, submicron and supermicron aerosol particles, respectively) and xylose (means= 16; 15 and 15 mol%) still prevailed over the relative monosaccharide pattern, the contribution of galactose (means= 6; 3 and 3 mol%) was strongly reduced, both in fog and aerosol particles. On the other hand, the ratio of muramic acid was



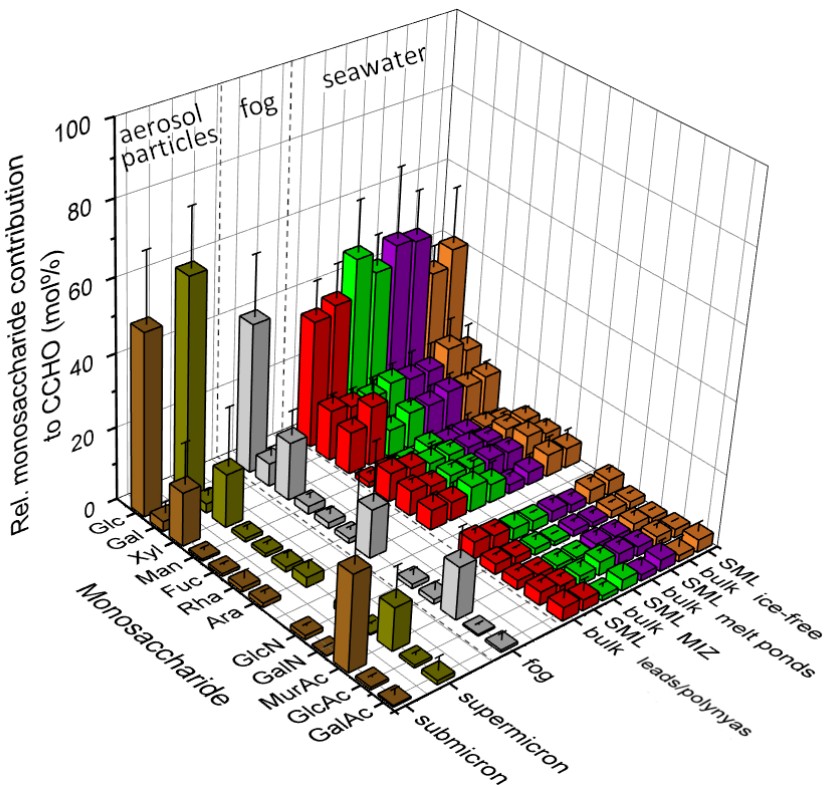

**Figure 7.** Relative monosaccharide composition of combined carbohydrates (CCHO) after acid hydrolysis in sub-/ supermicron aerosol particles, fog water, bulk and SML samples from the leads and polynyas within the pack ice, the MIZ, the ice-free ocean and young and aged melt ponds. The 3D bar chart shows the averages and standard deviations of the relative contributions. Glc: glucose, Gal: galactose, Xyl: xylose, Man: mannose, Fuc: fucose, Ara: arabinose, GlcN: glucosamine, GalN: galactosamine, MurAc: muramic acid, GlcAc: glucuronic acid, GalAc: galacturonic acid.

strongly elevated in aerosol particles (means= 12 and 26 mol%) and fog water (mean= 14 mol%) in
comparison to the oceanic samples (means= 0.9–2.6 mol%). These differences of the relative
monosaccharide contributions to CCHO among the seawater and the atmospheric samples described
within this study are in good agreement with the sea-air transfer investigations conducted in the
Southern Ocean at the western Antarctic peninsula (Zeppenfeld et al., 2021a). Consequently, the
occurring phenomenon might be independent from the sampling location and could be explained by
three possible atmospheric processes, such as (1) a chemo-selective sea-air transfer of certain oligo-
or polysaccharides over others, (2) an atmospheric transformation due to abiotic chemical reactions
or (3) an atmospheric transformation due to microbiological activities. Among these possible
pathways, Zeppenfeld et al. (2021) presumed the secondary atmospheric transformation caused by
microbiological metabolism as the most probable or at least most dominant one supported by the
prevalence of muramic acid, an amino sugar acid naturally occurring in bacterial cell walls (Mimura



and Romano, 1985; Sud and Tyler, 1964), and the very selective absence of certain monosaccharides
in the CCHO$_{aer}$ in aerosol particles as it was observed in this Arctic study as well.
***Formation of combined arabinose in fog.*** A comparison of the monosaccharide composition of aerosol
particles and fog water showed great similarity regarding dominant contributions from glucose, xylose
and muramic acid. It seems plausible that the fog water droplets contained the same inorganic and
organic compounds found in the SSA particles assuming that SSA particles activated the formation of
fog droplets as CCN due to their rather large diameters and high hygroscopicity. Apart from that,
however, a significant difference was observed in the increased relative contribution of arabinose in
fog (mean= 13 mol%) compared to aerosol particles (means= 1.2 and 2.7 mol%) indicating a formation
of arabinose in the liquid phase. During a marine microcosm experiment performed by Hasenecz et al.
(2020), a strong link was observed between the release of arabinose-containing polysaccharides in
form of EPS and the presence of heterotrophic bacteria and stressed phytoplankton. Furthermore, a
strain of the psychrotolerant marine bacterium *Pseudoalteromonas* sp. has been shown to produce
EPS mainly composed from glucose, arabinose and xylose (Casillo et al., 2018; Qin et al., 2007).
Consequently, the release of arabinose-containing EPS in fog could be a plausible protection
mechanism of microorganisms contained within a droplet against freezing damage under low Arctic
temperatures.
***Indication for microbial activities in the atmosphere.*** Intact bacterial cells at atmospheric
concentrations between $5 \times 10^2$ and $8 \times 10^4$ cells m$^{-3}$ for remote marine and ice-covered regions (Šantl-
Temkiv et al., 2018; Mayol et al., 2017), cell-bound and free enzymes have been detected in ambient
and nascent marine super- and submicron aerosol particles during several field and tank studies (Aller
et al., 2005; Hasenecz et al., 2020; Malfatti et al., 2019; Marks et al., 2001; Rastelli et al., 2017; Šantl-
Temkiv et al., 2020; Uetake et al., 2020). For surviving in this hostile environment, some of these
microbes have developed a remarkable resilience towards extreme environmental stressors, such as
high UV radiation, radical exposure, changing osmolarity, freezing temperatures and desiccation. As
survival strategies could serve the selective enzymatic consumption of airborne labile carbohydrates
explaining the here observed loss of galactose and the persistence of xylose, the formation of
protecting biofilm from EPS, carotenoid pigmentation or the formation of own precipitating
hydrometeors by enabling condensation on a surface as a CCN or freezing by IN active surfaces to
reduce their atmospheric residence time (Delort et al., 2010; Matulová et al., 2014; Šantl-Temkiv et
al., 2020). Consequently, an enzymatic transformation might serve as a plausible explanation for the
selective removal of certain monosaccharides within CCHO$_{aer}$ and CCHO$_{fog}$ observed here. However,
the survival and the metabolic activity of microorganisms is restricted by the presence of water (Ervens
and Amato, 2020; Haddrell and Thomas, 2017) identifying liquid hydrometeors or fresh SSA as the





most biologically active atmospheric hotspots. In contrast to most of the ambient aerosol particles, fog
droplets provide enough water essential for bacterial activities. However, they might freeze under
Arctic sub-zero temperatures possibly causing damage to the microbial cells, which might explain an
in-situ formation of a protecting biofilm from arabinose-containing EPS. In a previous Arctic study,
Orellana et al. (2011) readily detected microgels in aerosol particles, cloud and fog water most likely
emitted from the surface water and the SML via bubble bursting. Indications for an in-situ generation
of marine microgels in fog water as an additional source to the primary release from the ocean by
bubble bursting have been observed by van Pinxteren et al. (2022) in the tropical Atlantic Ocean.
The selective sea-air transfer of certain carbohydrates over others and the abiotic degradation as
further possible pathways to the biotic transformation of marine $CCHO_{aer}$ have been discussed in detail
in Zeppenfeld et al. (2021), but do not appear, based on the current state of knowledge, as likely
explanations of the very selective CCHO degradation and formation of other CCHO observed here.
More future lab and mesocosm experiments are required to elucidate the contribution of each of these
processes. Finally, the similarity of the carbohydrate composition of fog water and aerosol particles,
both two atmospheric compartments collected with different instrumentation, allows to rule out
artefacts of the different sampling and extraction techniques as a reason for the observed differences
to the seawater.
***Depolymerization of CCHO to dFCHO, seawater versus atmosphere.*** Free glucose, by far the most
prevailing monosaccharide among $d$FCHO in seawater, ranged between 0.6 and 51 µg L$^{-1}$ during the
PS106 cruise in the bulk and the SML (Zeppenfeld et al., 2019a). Thus, $d$FCHO/CHO ratios, meaning the
contribution of sugar monomers to all marine carbohydrates measured in this study, varied between
1–14% with an average of 5±3%. Conversely, 86–99% (mean: 95±3) of carbohydrates in the bulk and
SML of ocean seawater and melt ponds were incorporated into an oligo- or polysaccharidic structure.
CCHO can be hydrolyzed to $d$FCHO either in an acidic environment or enzymatically by heterotrophic
bacteria (Arnosti, 2000; Panagiotopoulos and Sempéré, 2005). Seawater from the Arctic Ocean is
slightly alkaline with reported pH values between 7.98 and 8.49 (Rérolle et al., 2016; Tynan et al.,
2016), while the pH of melt pond water has been observed to be more variable from mildly acidic (6.1)
to more alkaline (10.8) (Bates et al., 2014). In agreement with previous findings, the oceanic surface
seawater (pH: 7.98–8.66), including the samples from the MIZ, ice-free ocean and open
leads/polynyas, and the melt pond samples (pH: 7.26–8.62) were slightly alkaline in this study.
Consequently, it is more plausible that the depolymerization of CCHO in seawater can be ascribed to
bacterial activities rather than acid hydrolysis. Since $d$FCHO are readily resorbed by heterotrophic
bacteria with high turnover rates (Ittekkot et al., 1981; Kirchman et al., 2001), concentrations of these
monosaccharides are rather low in seawater.



In contrast, in aerosol particles, higher $d$FCHO/CHO ratios up to 35% occurred in some selected samples, which is much higher than in seawater, suggesting that CCHO might be depolymerized in the atmosphere. SSA particles are known to significantly acidify within minutes after their release due to the uptake of acidic gases, atmospheric aging reactions with sulfuric dioxide and water loss (Angle et al., 2021). In this context, the surface-to-volume ratio determines the efficiency of the acidification effect, which means that it is most pronounced for submicron SSA particles with reported pH values of 1.5–2.6 within a few minutes in a tank study (Angle et al., 2021), and less pronounced for supermicron SSA particles or cloud droplets (Angle et al., 2022). Consequently, it is conceivable that an acid hydrolysis of $CCHO_{aer}$ to monomeric $d$FCHO$_{aer}$ occurs at the surface or within the bulk of SSA aerosol particles leading to quick atmospheric aging. However, due to analytical constraints, such as the limits of detections (LODs) of the methodology, the $d$FCHO in size-resolved aerosol particles could not be detected in all samples and the data availability is not strong enough to draw more conclusions for aerosol particles.

In fog, where LODs did not represent an issue due to the high concentrations, $d$FCHO/CHO ratios higher than to seawater occurred (1–60%, mean: 27±16%) as well. The monosaccharide composition of $d$FCHO$_{fog}$ was dominated by glucose, arabinose, fructose and xylose with small contributions from glucosamine, galactose, mannose, rhamnose and fucose. Consequently, the monosaccharide composition of $d$FCHO$_{fog}$ was quite similar to CCHO$_{fog}$, just with the difference that fructose was detected in $d$FCHO$_{fog}$, but not in CCHO$_{fog}$, which is due to the low stability of fructose towards the analytical preparation procedure for the analysis of CCHO (Panagiotopoulos and Sempéré, 2005) and should, hence, not find further considerations. In this study, pH values of fog water ranged between 5.7 and 6.8, which is 1–2 magnitudes more acidic than in seawater. Polysaccharides are known to depolymerize due to acid hydrolysis, especially at elevated temperatures. The pH-stability can be largely variable among the different polysaccharides; however, we are not aware of studies that have shown such fast depolymerizations, in the sense of time scales relevant for atmospheric lifetime of aerosol particles, at such mildly acid conditions and low temperatures as those of the Arctic atmosphere. Furthermore, there was no significant correlation between the pH and the $d$FCHO/CHO of these cloud samples. Consequently, there are no indications that the majority of CCHO was hydrolyzed inside the cloud droplets, however it might be conceivable that hydrolysis had readily occurred within the non-activated SSA particle where pH values were much lower.

Besides an acid hydrolysis induced by quick atmospheric acidification of SSA particles, atmospheric radicals, such as OH (Trueblood et al., 2019), or photolytic cleavages of glycosidic bonds (Kubota et al., 1976) could have contributed to the degradation of atmospheric CCHO to monomeric $d$FCHO in SSA and marine fog. For these processes, however, still hardly any systematic lab studies have been



conducted for the plurality of marine polysaccharides, which makes a classification of the meaning of
these processes difficult. A preferred sea-air transfer of $d$FCHO over CCHO to explain this observation
seems unlikely based on the missing enrichment of neutral $d$FCHO in contrast to the high $EF_{aer}$ of CCHO
shown in tank studies (Hasenecz et al., 2019, 2020). Finally, a microbial depolymerization of CCHO by
extracellular enzymes in fog cannot be entirely ruled out considering that the activity of some
polysaccharide-degrading enzymes, such as α- and β-glucosidase, have been found to accelerate in
seawater with increasing acidity (Piontek et al., 2010). However, this finding was conducted for a pH
range only 0.3 pH units lower than the typical pH of seawater and it is not sure, if this finding can be
transferred to the more acid conditions in aerosol particles and fog water.
***Several aging processes in the atmosphere.*** We observed significant changes between the chemical
composition of marine carbohydrates in the surface seawater, including the bulk and SML, and
atmospheric carbohydrates, including aerosol particles and fog. Based on the changing
monosaccharide composition pattern of CCHO with selective degradation and formation of specific
monosaccharides within CCHO, we conclude microbial or enzymatic activities within the aerosol
particles of fog droplets. Furthermore, the increasing contribution of $d$FCHO to the total carbohydrate
pool in fog and aerosol particles might be attributed to a hydrolytic cleavage of the glycosidic linkages
between monosaccharide unites within the oligo-and polysaccharides after a quick atmospheric
acidification of SSA particles. Consequently, atmospheric carbohydrates experience quick atmospheric
aging, potentially due to both biological and abiotic processes, after their release from the ocean.
Possibly, this could affect the CCN and INP properties of marine carbohydrates and hence the
formation and properties of clouds.



### 3.6 Perspective assessment of CCHO via bio-optical parameters

The absorption of phytoplankton ($a_{ph}$) and CDOM ($a_{CDOM}$) are bio-optical parameters providing additional information about the chemical and microbiological history of the water masses within the particulate and dissolved phase, respectively. They can be measured on discrete water samples and can also be assessed as products from satellites (Lefering et al., 2017; Matsuoka et al., 2012, 2013; Röttgers et al., 2016). Here we tested, if $a_{ph}$ or CDOM parameters correlate with CCHO in seawater to potentially enable the remote-sensing approximation of marine CCHO in seawater and potentially in the atmosphere.

***Good assessment of CCHO in seawater via $a_{ph}$440.*** $a_{ph}$440 derived from the phytoplankton absorption spectrum is directly related to the biomarker TChl-*a* indicating phytoplankton biomass (Bricaud et al., 2004; Phongphattarawat, 2016). The advantage of using $a_{ph}$440 over pigment data, including TChl-*a* from full high-performance liquid chromatography (HPLC) analysis (e.g. Barlow et al., 1997; Taylor et al., 2011), is the lower need of sample volume for the analysis. This allows the determination of values in the SML samples as well (Zäncker et al., 2017), which are laborious to collect and therefore limited in availability. In this study, $a_{ph}$440 strongly correlated with *p*CCHO (R=0.90, p<0.001) in bulk and SML samples (**Figure SI 4**) showing a direct link with fresh phytoplankton biomass production. A similar link has been described before for TChl-*a* and *p*CCHO in the photic layer of the Ross Sea (Fabiano et al., 1993), in the ocean west of the Antarctic peninsula (Zeppenfeld et al., 2021a) and between TChl-*a* and the particulate form of laminarin, an algal polysaccharide, in Arctic and Atlantic water samples (Becker et al., 2020). *d*CCHO showed a good, but weaker correlation with $a_{ph}$440 (R=0.66, p<0.001) than *p*CCHO. This finding supports the assumption that *p*CCHO are rather freshly produced by local autotrophs, while the link between *d*CCHO with their primary production was already contorted by subsequent transformation processes resulting in a more recalcitrant, long-lived mix of macromolecules (Goldberg et al., 2011; Hansell, 2013; Keene et al., 2017). Nevertheless, CCHO, the sum from *d*CCHO and *p*CCHO, showed a high correlation with $a_{ph}$440 (R=0.84, p<0.001) leading to the conclusion that this bio-optical parameter derived from the $a_{ph}(\lambda)$ spectrum is suitable to assess the total amount of CCHO in the surface seawater of the different sea-ice related sea surface compartments of the Arctic.

***Good assessment of CCHO in seawater via $a_{CDOM}$350.*** In this study, high correlations were observed between *d*CCHO and $a_{CDOM}$350 (R=0.66, p<0.001, **Figure SI 5a**), and weaker correlations between *d*CCHO and $a_{CDOM}$443 (R=0.53, p<0.001, **Figure SI 5b**). The better correlation at $\lambda$=350 nm compared to 443 nm can be explained by the fact that $a_{CDOM}$ exponentially decreases with wavelength. While absorption by CDOM is higher at $\lambda$=350 nm, it is much closer to the method detection limit at $\lambda$=443 nm



and is therefore more error-prone. However, with current satellite products only $a_{CDOM}$ at 440 nm can
be retrieved.
Previous studies reported strong correlations between $a_{CDOM}350$ and dissolved organic carbon (DOC)
in Arctic seawater (Gonçalves-Araujo et al., 2015; Spencer et al., 2009; Stedmon et al., 2011; Walker
et al., 2013). Consequently, it is conceivable that $d$CCHO, an important constituent of DOC, shows good
correlations as well. Surprisingly, the correlation between CCHO (sum of $d$CCHO and $p$CCHO) and
$a_{CDOM}350$ was strongest (R=0.85, p<0.001, **Figure SI 5c**), indicating that CDOM retrieval from high-
resolution satellite data could allow a good approximation of CCHO in Arctic seawater.



## 4. Summary and Atmospheric Implications

We studied the sea-air transfer of marine carbohydrates from field samples collected in the Arctic during the PS106 campaign from May to July 2017. Large differences of absolute CCHO concentrations and SML enrichments were observed among the different sea-ice related sea surface compartments (leads/polynyas within the pack ice, ice-free ocean, MIZ, melt ponds). $CCHO_{aer}$ were detected in the sub- and supermicron aerosol particles with indications for primary emissions from the sea through bubble bursting, though the correlations with the SSA tracer $Na^+$ and wind speed were possibly reduced due to the presence of sea ice influencing the wind-induced SSA emission mechanisms. Atmospheric CCHO concentrations in fog strongly exceeded those of the aerosol particles, which might be due to a phenomenon called fog scavenging and partly the comparability of the different sampling approaches for fog and size-resolved aerosol particles. A large enrichment for CCHO in aerosol and fog relative to seawater was observed, the extend of which varied on the type of sea-ice related sea surface compartment assumed as the oceanic source. We observed a subsequent atmospheric aging of CCHO in the atmosphere, both in aerosol particles and fog, noticed by the selective loss and formation of certain monosaccharide units within CCHO suggesting selective enzymatic/microbial activities, and a depolymerization of CCHO to $d$FCHO, most measurable in fog water and likely due to abiotic degradation, e.g. acid hydrolysis. CCHO correlated well with bio-optical parameters, such as $a_{ph}440$ from phytoplankton absorption and $a_{CDOM}350$. These parameters can be measured via remote sensing and may allow the retrieval of CCHO from satellite data, which potentially will enable an accurate modelling of atmospheric CCHO concentrations as soon as all emission and atmospheric aging processes are sufficiently understood. In a nutshell, this study shows that the Arctic is a complex environment, where the diversity of sea-ice related sea surface compartments needs to be considered as primary sources of marine CCHO or other organic compounds, and where these molecules can be transformed after their primary sea-air transfer by biological and abiotic processes in the atmosphere.

Marine carbohydrates are assumed to impact cloud properties by acting as CCN and INP (Leck et al., 2013; Orellana et al., 2011; van Pinxteren et al., 2022). Studying the chemical identity of those atmospheric nucleation particles, their emission mechanisms and their transformation due to atmospheric aging can strongly improve the understanding of the cloud formation in the Arctic, cloud microphysical properties, the radiation budget, cryosphere-ocean-atmosphere interactions and eventually feedback mechanisms in the frame of Arctic amplification. It can be assumed that within the warming Arctic, where sea ice extent is continuously shrinking, the MIZ area will expand (Strong and Rigor, 2013) and the number of biologically-active melt ponds will increase during the summer season in the next years. These new MIZ regions and melt ponds could potentially produce more marine carbohydrates than the ice-free ocean or open leads within the pack ice leading to enhanced



CCN and INP populations in the Arctic atmosphere serving as a still not well-explored feedback
mechanism within Arctic amplification.
***Data availability.*** All data will be made available on the public repository PANGAEA.
***Author contribution.*** SZ wrote the manuscript with contributions from MvP, MH, MZ, AB and HH. SZ,
MvP and MH collected the field samples during the PS106 campaign. SZ performed the laboratory
carbohydrate analysis and statistical evaluation. MZ and AB assessed the bio-optical parameters. All
co-authors proofread and commented the manuscript.
***Competing interests.*** The authors declare that they have no conflict of interest.
***Acknowledgements.*** We gratefully acknowledge the funding by the Deutsche Forschungsgemeinschaft
(DFG, German Research Foundation, Projektnummer 268020496−TRR 172) within the Transregional
Collaborative Research Center "ArctiC Amplification: Climate Relevant Atmospheric and SurfaCe
Processes, and Feedback Mechanisms (AC)[3]". This research has been supported by the DFG SPP 1158,
grant number 424326801 by enabling the access to melt pond data. We thank Andreas Macke and
Hauke Flores, chief scientists for the RV *Polarstern* cruises PS106.1 and PS106.2 (expedition grant
number AWI_PS106_00), and the captain and the crew of RV *Polarstern* for their support during the
expedition from May to July 2017. We thank Andrea Haudek and Hartmut Haudek for the development
and construction of the conditioning tube and the wind control system connected to the Berner
impactor. We thank Anett Dietze, Susanne Fuchs and Anke Rödger for the mass, inorganic ion and
OC/EC measurements. We acknowledge René Rabe and Sonja Wiegmann for supporting the
preparation of PS106 chemical equipment and optical instrumentation, respectively, and Yangyang Liu
for introducing the optical measurement procedure before PS106.
***Financial support.*** This research has been supported by the Deutsche Forschungsgemeinschaft (DFG,
German Research Foundation, Projektnummer 268020496−TRR 172) within the Transregional
Collaborative Research Center "ArctiC Amplification: Climate Relevant Atmospheric and SurfaCe
Processes, and Feedback Mechanisms (AC)[3]" in subprojects B04 and C03.
**List of abbreviations**
$a_{CDOM}$ absorption coefficient by colored dissolved organic carbon
aer aerosol particles
$a_{NAP}$ absorption coefficient by non-algal particles
ANOVA Analysis of Variance
$a_p$ absorption coefficient by total particles
$a_{ph}$ absorption coefficient by phytoplankton
Ara arabinose
atmos atmospheric concentrations
C-CCHO carbon contained within the combined carbohydrate
CCHO combined carbohydrates
CCN cloud condensation nuclei



| 917 | CDOM | colored dissolved organic matter |
| 918 | CHO | carbohydrates |
| 919 | *d*CCHO | dissolved combined carbohydrates |
| 920 | *d*FCHO | dissolved free carbohydrates |
| 921 | EF | enrichment factor |
| 922 | EPS | exopolymeric substances |
| 923 | ERDDAP | Environmental Research Division's Data Access Program |
| 924 | FAA | free amino acids |
| 925 | Fru | fructose |
| 926 | Fuc | fucose |
| 927 | Gal | galactose |
| 928 | GalN | galactosamine |
| 929 | GalAc | galacturonic acid |
| 930 | Glc | glucose |
| 931 | GlcAc | glucuronic acid |
| 932 | GlcN | glucosamine |
| 933 | HPAEC-PAD | high-performance anion-exchange chromatography with pulsed amperometric detection |
| 934 | HPLC | high-performance liquid chromatography |
| 935 | INP | ice nucleating particles |
| 936 | LWCC | liquid waveguide capillary cell |
| 937 | Man | mannose |
| 938 | MIZ | marginal ice zone |
| 939 | MurAc | muramic acid |
| 940 | Na$^+$ | sodium ion |
| 941 | NOAA | National Oceanic and Atmospheric Administration |
| 942 | OC | organic carbon |
| 943 | OM | organic matter |
| 944 | PAB | particulate absorption |
| 945 | *p*CCHO | particulate combined carbohydrates |
| 946 | PM | particulate matter |
| 947 | Rha | rhamnose |
| 948 | SML | sea surface microlayer |
| 949 | SSA | sea spray aerosol |
| 950 | sub | submicron |
| 951 | super | supermicron |
| 952 | TChl-*a* | total chlorophyll *a* |
| 953 | TEP | transparent exopolymer particles |
| 954 | QFT-ICAM | quantitative filtration technique with an integrative-cavity absorption meter setup |
| 955 | Xyl | xylose |



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
