# Peer review of "Marine Carbohydrates in Arctic Aerosol Particles and Fog – Diversity"

_EGUsphere, 2023_

## Author Comment (AC1)

Answers to the Referees' comments regarding the manuscript:

**Marine Carbohydrates in Arctic Aerosol Particles and Fog – Diversity of Oceanic Sources and Atmospheric Transformations**

Sebastian Zeppenfeld1, Manuela van Pinxteren1, Markus Hartmann2, Moritz Zeising3, Astrid Bracher3,4, and Hartmut Herrmann1

1 Atmospheric Chemistry Department (ACD), Leibniz-Institute for Tropospheric Research (TROPOS), Leipzig, Germany

2 Atmospheric Microphysics (AMP), Leibniz-Institute for Tropospheric Research (TROPOS), Leipzig, Germany

3 Alfred-Wegener-Institute Helmholtz Centre for Polar and Marine Research, Bremerhaven, Germany 4 Institute of Environmental Physics, University of Bremen, Bremen, Germany

\*Correspondence to: Hartmut Herrmann (herrmann@tropos.de)

MS No.: egusphere-2023-1607 MS type: Research article

We thank both reviewers for the evaluation of our manuscript. In this document, all of their constructive comments have been answered thoroughly. The referees' comments are marked blue, our replies black, and changed text in the manuscript green. The given line numbers of changed sentences are referring to the new lines in the revised manuscript.

**Reviewer: 1**

Zeppenfeld et al. investigated the transfer and composition of marine carbohydrates in the Arctic atmosphere based on samples collected during the PS106 campaign in May-June 2017. Various sea surface compartments showed significant differences in carbohydrate concentrations and enrichment factors. Atmospheric aging of carbohydrates was observed, influenced by enzymatic/microbial activities and abiotic degradation. These findings have implications for cloud properties and interactions between cryosphere, ocean, and atmosphere in the Arctic, with potential feedback mechanisms. The manuscript is well written and very thorough. I only have some general and specific comments below that the authors should address prior to publication in EGUsphere.

**Authors:**

We thank the reviewer for the constructive suggestions and carefully revised the manuscript accordingly.

**General comments:**

Abstract: Should contain 1-3 opening sentences that discuss the motivation and important of this study, and briefly define the region (Fram Strait) and time of year (May-Jun 2017).

Authors: We agree with the reviewer that this missing information should be added to the abstract. The changed text now reads: 'Carbohydrates, produced and released by marine microorganisms in the ocean, enter the atmosphere as part of sea spray aerosol (SSA) and can influence fog and cloud microphysics by acting as cloud condensation nuclei (CCN) or ice nucleating particles (INP). Particularly in the remote Arctic region, significant knowledge gaps persist about the sources, the sea-to-air transfer mechanisms, atmospheric concentrations, and processing of this substantial organic group. In this study, we present the results of a ship-based field study conducted from May to July 2017 in the Fram Strait, Barents Sea and central Arctic Ocean about the sea-air transfer of marine combined carbohydrates (CCHO) from concerted measurements of the bulk seawater, the sea surface microlayer (SML), aerosol particles and fog.' (Lines 19-26)

"Aged" melt ponds: How was the age of the melt ponds determined? It comes up a few times throughout the manuscript but it is not clear how they are defined from newly-formed ponds. Usually this time of year corresponds to relatively new melt ponds, while older ponds are present during the end of summer (e.g., when they form at their deepest depths and even drain).

Authors: We categorized melt ponds as young or aged based on their visual characteristics. Samples from young melt ponds, which were smaller, bluish, and clear, were collected in mid-June. In contrast, aged melt pond samples, taken at the end of June or July, were larger, exhibited a darker blue to greenish hue, and appeared more turbid due to visible particulates and microalgae. While our two-month participation in the PS106 campaign didn't encompass the entire melt pond evolution until late summer, the rapid shift in their biological and chemical properties was evident. We've detailed our categorization criteria for young and aged melt ponds in the experimental section of our manuscript: 'Using visual characteristics, melt ponds were categorized as young (small, bluish, clear) or aged (larger, darker blue to greenish, and turbid with particulates and microalgae).' (Lines 157-159)

Was melt pond depths measured? If so, was there any relationship with the presence and production of CCHO in the water and air? That would somewhat get to if there is a relationship with newer versus older ponds.

Authors: We agree that there is a high potential value in exploring the relationship between melt pond depth and CCHO in the melt ponds and in the air. Indeed, we did measure melt pond depths in this study

and found that younger melt ponds with lower CCHO concentrations ranged from 20-40 cm deep, while aged ones with higher CCHO concentrations varied from 40 cm to open-bottomed. However, given our limited dataset of only six melt pond samples, we cannot conclude on correlations between melt pond depth, age, and CCHO concentration these few data points. It's indeed an intriguing research question worth pursuing in future dedicated studies.

**This is just a minor comment, but 'chapters' is used a few times and I think the authors mean 'sections'.**

Authors: Thank you for pointing out the incorrect use of the word 'chapters'. We have replaced it with 'sections', accordingly.

**Specific comments:**

Line 23: Define the EF with subscripts after first mention, then just include the values in the parentheses in the rest of the sentence.

Authors: We thank the reviewer for this suggestion. The changed sentence within the abstract now reads: 'Enrichment factors in the SML (EFSML) relative to the bulk water were very variable in the dissolved (0.4–16) and particulate (0.4–49) phases with highest values in the MIZ and aged melt ponds.' (Lines 29-31)

**Line 28: Does 'CCHO\_fog, atmos' refer to interstitial aerosol present during fog events?**

Authors: Interstitial aerosol particles refer to the non-activated particles found between fog droplets. CCHOfog,atmos measured here, represents the atmospheric concentrations in fog droplets, hence, not in the interstitial aerosols.

Fog samples were analyzed as liquid samples resulting in concentrations in ( $\mu$ g CCHO) / (L fog water). However, atmospheric concentrations in (ng CCHO) / (m3 air) are required to compare the concentrations in fog with ambient aerosol particles. This conversion necessitates liquid water content (LWC) values. The more detailed explanation on this topic is given in section 3.3. A comprehensive discussion is presented in section 3.3, where we've added clarity for the readers. The changed sentence now reads: 'Atmospheric concentrations of these chemical constituents in fog droplets (indicated by the index 'fog,atmos') can be calculated under consideration of the liquid water content (LWC) during the fog events.' (Lines 529-531)

Line 40: I think "boreal" is the incorrect word to use here since the focus is on the Arctic Ocean and not the subpolar climate zone.

**Authors: We agree that 'boreal' is the incorrect word. We replaced it with 'Northern Hemisphere'. The changed sentence now reads:**

'Sea spray aerosol (SSA) represents one of the major aerosol species in the lower troposphere over the remote Arctic Ocean, particularly during the spring and summer months in the Northern Hemisphere (Chi et al., 2015; Hara et al., 2003; Kirpes et al., 2018; May et al., 2016).' (Lines 44-46)

Lines 40-41: Kerri Pratt's group has done quite a bit of work on SSA in the Arctic. It would be useful to check out and cite those papers (e.g., Kirpes et al. (2018, 2019), May et al. (2016)).

Authors: We agree that these papers are suitable for citation here. The changed text now reads: 'Sea spray aerosol (SSA) represents one of the major aerosol species in the lower troposphere over the remote Arctic

Ocean, particularly during the spring and summer months in the Northern Hemisphere (Chi et al., 2015; Hara et al., 2003; Kirpes et al., 2018; May et al., 2016).' (Lines 44-46)

**Lines 41-45: Probably should add a few recent, key references here, even though some of these elude to blowing snow as the source of SSA - Chen et al. (2022), Huang and Jaegle (2017).**

Authors: Thanks for recommending these references on the subject of blowing snow, which we found more appropriate to cite later in our manuscript. Furthermore, we decided to still add the recently published paper by Gong et al. (2023) on blowing snow in the Arctic. The changed text now reads: 'Finally, blowing snow over the sea ice could serve as an additional source of atmospheric Na+aer when a certain air-temperature-dependent wind speed threshold is exceeded. (Chen et al., 2022; Gong et al., 2023; Huang and Jaeglé, 2017; Yang et al., 2008).' (Lines 456-458)

**Line 47: How does 'primary marine aerosol particles' differ from SSA? Maybe just stick to one, common term.**

Authors: We used both terms as synonyms, and we agree with the reviewer that sticking to one is better for the understandability of the manuscript. The changed text now reads: 'Notably in the Arctic, one of the regions most affected by global warming, there is still a lack of knowledge about the relationship between the formation and evolution of clouds and specific chemical properties of SSA particles (Wendisch et al., 2023).' (Lines 51-53)

**Lines 64-68: Is another difference between dissolved and particulate CCHO due to size, whether they are single carbohydrates or agglomerates of them, respectively? That would make sense to me on a physical basis.**

Authors: We agree that this makes sense on a physical basis. Individual carbohydrate molecules are logically classified under the dissolved CCHO fraction, while larger clusters unquestionably belong to the particulate CCHO category. However, the transition from truly dissolved molecules via colloidal nanogels to microscopic particles in seawater is continuous along the whole size range (Verdugo et al., 2004). Given the challenge to resolve this with conventional analytical tools, the demarcation between dissolved and particulate CCHO is usually performed via a filtration at pore sizes varying between 0.2 and 1.0  $\mu$ m. In this study, we set the threshold at 0.2  $\mu$ m. We clearified this definition of 'dissolved' and 'particulate' in the introduction of our manuscript. The changed text now reads: 'In seawater, most carbohydrates appear as linear or branched oligo- and polysaccharides, commonly referred to as combined carbohydrates (CCHO). They can be found in both dissolved (dCCHO) and particulate (pCCHO) phases, distinguished operationally by a 0.2  $\mu$ m filtration.' (Lines 60-63)

Lines 73-73: Should also cite Alpert et al. (2022). Include Alpert et al. (2022) on lines 870-871 as well.

Authors: Done.

Line 116: Leads can often be wider than several meters (sometimes 10s), and hundreds of meters long.

**Authors: We agree with the reviewer. The changed text now reads:**

'These encompass the open leads - sea ice fractures with variable widths ranging from several to hundreds of meters - and polynyas, which are larger, more persistent areas of open water within the pack ice.' (Lines 121-123)

Line 149: How was the SML thickness measured/calculated? Was it based on the total mL of SML collected and the collection speed? If so, is that representative of the wider region or is the thickness variable locally?

**Authors:**

**-calculation of the SML thickness**

The calculation of the SML thickness was performed as described by Cunliffe and Wurl (2014) and is based on the total mL of SML collected, the area of the glass plate and the number of manual dips that were performed for collecting the respective volume of SML. We added this information to the main text of the manuscript. The changed text now reads: 'The average thickness of the SML collected during this field study was 76±10  $\mu$ m, which was calculated based on the volume of the SML sample collected, the area of the immersed glass plate and the number of dips as described by Cunliffe and Wurl (2014).' (Lines 165-168)

**-based on collection speed?**

The collection speed could influence the sample volume, but is not considered in the calculation of the SML thickness. To keep this influence as small as possible and to maintain the comparability among all samples, we performed the sampling withdrawal at a controlled speed of approximately 15 cm s-1 for all samples.

**-representative of wider region?**

The resulting SML thickness is dominated by several factors, such as the sampling method (glass plate technique), the stickiness of the SML film, wind speed/wave state, water temperature or biological activity (Cunliffe and Wurl, 2014; van Pinxteren et al., 2017). Even though we found quite homogeneous SML thicknesses by applying the SML technique at similarly calm wave states and cold water temperatures in the high Arctic during the PS106 campaign, it is plausible that these results do not necessarily represent the entire Arctic or Arctic region. More research is required in this field.

**Line 151: What is meant by 'closed' melt ponds? Ice lids on the ponds?**

Authors: We use 'closed' melt ponds to state that melt ponds were closed at the bottom surrounded by sea ice and consequently were not in direct contact with the sea surface water below. Since closed melt ponds showed a depth of 20-40 cm depth during this study, bulk sampling at 1 m depth was not possible within these 'bottom closed melt ponds'. It is not referring to ice lids on the ponds as assumed by the reviewer. To clarify this to the readers, we replaced 'closed melt ponds' with 'bottom closed melt ponds'. The changed sentence now reads: 'The corresponding bulk water was taken from a defined depth of 1 m into LDPE bottles attached to a telescopic rod, except in the bottom closed melt ponds where it was scooped from the bottom at approximately 20–40 cm depth.' (Lines 170-172)

**Line 157: I assume the 0.2 um filtration was to remove PBAP?**

Authors: We believe the reviewer uses 'PBAP' to abbreviate 'primary biological aerosol particles'. However, this section refers to the pretreatment of seawater samples prior to storage in a freezer, and not to aerosol particles.

Based on our experience in marine field studies, we preserve the dissolved ( $<0.2\mu$ m) and the particulate ( $>0.2\mu$ m) fractions of the seawater samples separated from each other before freezing to increase the chemical/microbiological stability of the samples and the reproducibility of the measurements. To make it clearer to the readers, we changed the text, which now reads: 'For later chemical analyses (inorganic ions,

pH, carbohydrates) 500–1000 mL of 0.2  $\mu$ m filtered water sample (dissolved fraction), 0.2  $\mu$ m polycarbonate filters (particulate fraction) and field blanks were stored at -20°C.' (Lines 178-180)

Lines 281-282: Should say SSA here instead or 'primary marine aerosol particles'. And also not that it is not just carbohydrates, but also microbial cells and fragments.

Authors: We changed this sentence accordingly, which now reads: 'The sources of SSA particles, and hence of atmospheric marine carbohydrates, microbial cells and fragments, in the Arctic are diverse and influenced by the prevailing sea ice conditions.' (Lines 306-307)

Fig SI2: Since there is a bit of discussion on the results in this figure, the authors should strongly consider moving it to the main manuscript.

Authors: We have located 'Figure SI 2' from the supplementary information to the main manuscript, where it is now referred to as 'Figure 3'.

Lines 453-456: Could this difference also be because the SO/Antarctic pack ice does not have melt ponds that develop in the austral summer? Would make sense why Na+ would correlate better with CCHO in the SO than the Arctic.

Authors: Yes, we agree with the reviewer. Our intent was to convey this point in the original sentence, but it seems the clarity was lacking. To ensure this aspect is more transparent to the reader, we have expanded upon the sentence. The changed text now reads: 'Unlike the study conducted in the Southern Ocean (Zeppenfeld et al., 2021a), CCHOaer,PM10 in this study showed no significant correlations with Na+aer,PM10 (R=0.24, p>0.1, **Figure 3b**) or wind speed (R=0.26, p>0.1, **Figure 3c**). The presence of sea ice resulting in melt ponds and MIZ regions, and the interplay of multiple emission mechanisms in the Arctic, as discussed earlier, could account for this complexity.' (Lines 497-500)

Lines 478-480: When and where were these samples collected? That info should be included in the methods.

Authors: We added this information about snow sampling to the Experimental, section 2.1. The added sentence now reads: 'Whenever melt pond sampling took place, snow samples were collected from the ice floe surface roughly 10 m away from the melt pond.' (Lines 172-173)

Lines 507-510: The authors should provide some concrete evidence (i.e., values and citation(s)) from previous Arctic fog studies.

Authors: We added two references (Herrmann et al., 2015; Kumai, 1973) to give concrete evidence. Furthermore, we gave more detail about the spatial variability of the LWC values within fog samples collected over the North Sea/Norwegian Sea and the Arctic Ocean during PS106. The changed texts now read:

'This approach resulted in LWCs of 0.62 ±0.39 g m-3 for the fog collected over the North Sea and Norwegian Sea, and 0.10±0.09 g m-3 for the fog over the Arctic Ocean (Hartmann et al., 2021).' (Lines 533-535)

and

'This approach resulted in values representing rather the upper limit of LWC values typically reported for Arctic summer fog ( $0.001-0.17 \text{ g m}^{-3}$  (Kumai, 1973)) or sea fog ( $0.02-0.1 \text{ g m}^{-3}$  (Herrmann et al., 2015)), but

appear within a realistic range. Consequently, they are likely not responsible for the large difference between aerosol and fog concentrations of several orders of magnitude.' (Lines 553-558)

Lines 570-575: It is hard to tell quantitatively how much time air masses spent over the ice, ocean, etc. from Fig SI 3. How was this calculated? The authors should show these calculator results, either in a table or another SI figure.

**Authors:**

**-calculation**

We agree and provided additional details regarding the quantification of the residence time of air masses over specific surface features in the '2.8 Statistics, calculations, and visualization' section: 'Time-resolved back-trajectories and sea ice maps were combined using R to compute and visualize the air mass history regarding the sea-ice-related sea surface compartments that have been passed. As a result, relative residence times of the air masses over certain surface features (ice-free, MIZ, pack ice, land) 12 hours before sampling were calculated based on defined thresholds for sea ice concentration: less than 15% for ice-free ocean, 15-80% for MIZ and over 80% for pack ice. Based on the remote sensing data used, we did not distinguish between open leads and melt ponds as the air traversed the pack ice.' (Lines 281-287)

**-results in table or another SI figure**

These calculated results can be found in **Table SI 1**, which we have now referenced more prominently in the text for clarity: 'During four aerosol sampling periods (24/05/17-26/05/17; 26/05/17-29/05/17; 29/05/17-01/06/17; 19/06/17-25/06/17), air masses had predominantly passed over the ice-free ocean (45–100% of the 12 hours prior to sampling, as shown in **Table SI 1** & **Figure SI 2**). Interestingly, these periods exhibited the lowest CCHOaer,PM10/Na+aer,PM10 ratios (2×10-3–9×10-3, detailed in **Table SI 1**), values that are strikingly similar to those observed in the ice-free Southern Ocean.' (Lines 619-623)

Lines 603-606: It seems like these assumed sources were defined by the air mass analysis, but akin to my comment above, some additional methodology should be included. For example, was the 'ice-free ocean' source determined by each trajectory spending X percentage of time over the ice-free ocean south of the pack ice? What is the difference between 'marine' and 'ice-free ocean' in this case? Some thresholds for the relative amount of time each trajectory spent over each surface type, and some statistical analysis on these results would help.

**Authors:**

**-assumption of sources from air mass analysis and additional methodology**

The reviewer's feedback highlighted the lack of clarity and understandability in this section. While we did calculate the percentage of trajectory time over various sea surfaces as detailed in the previous section, we couldn't distinguish between the categories of 'young melt ponds', 'aged melt ponds', and 'open leads' on pack ice - sea-ice-related sea surface compartments with different chemical and biological characteristics. This challenge arises from their proximity to one another and the spatial uncertainties in high-resolution model computations. Moreover, we lacked knowledge on deposition rates, wind effects on wave propagation and bubble bursting within these different sea-ice-related sea surface compartments, and data regarding biological activity in individual melt ponds (elaborated in prior sections).

To account for these problems in the revised version we therefore refrained from calculating enrichment factors based on the percentage of time over these sea-ice-related sea surface compartments due to potential inaccuracies from these unknown variables.

Instead, the presented enrichment factors are hypothetical, assuming results based on a single source, overlooking the sea-ice compartment diversity. The primary takeaway is the necessity to consider all sea-ice-related compartments to avoid incorrect conclusions.

We've revised the manuscript to emphasize the hypothetical nature of this section and give more information on the methodology. The updated text is as follows: 'Due to missing information, including SSA emission fluxes from the four sea-ice-related compartments, aerosol deposition rates, biological activities in melt ponds, wind effects on wave propagation and bubble bursting, and the comparative importance of melt ponds versus open leads (which are in close proximity, making it difficult to resolve them in back-trajectory analyses) as SSA sources – we didn't perform calculations based on the back-trajectory history of each atmospheric sample. Instead, subsequent calculations for  $EF_{aer}$  and  $EF_{fog}$  employed a hypothetical approach, assessing the range of enrichment factors by considering only one of the four sea-ice-related compartments—represented by the corresponding median CCHObulk/Na+bulk ratios—as the only source, while excluding the others.' (Lines 651-659)

**and**

'It is important to note that EFs were most consistent with results from other CCHO sea-air transfer studies in the tank (Hasenecz et al., 2020) and the field (Zeppenfeld et al., 2021a), when aged melt ponds or the MIZ were considered as the only emission source. If leads/polynyas and the ice-free ocean were regarded as the only emission source, higher  $EF_{aer}$  and  $EF_{fog}$  values were obtained, and hence a possible overestimation of the mechanistic process of enrichment. As the results on back-trajectory calculations and sea ice maps demonstrated (**Table SI 1 & Figure SI 2**), most air masses were exposed to several of the sea-ice-related sea surface compartments before sampling. Consequently, none of the Arctic sea-icerelated sea surface compartments discussed above should be overlooked when discussing of sea-air transfer of organic substances.' (Lines 664-673)

**-difference between 'marine' and 'ice-free ocean'?**

We define 'marine source' as all sea-ice-related compartments discussed in this manuscript, with the 'ice-free ocean' being one of them. We concur with reviewer that the term 'marine' might be misleading in this context, so we have removed it. The changed text now reads: 'Lower atmospheric EFs were calculated when aged melt ponds ( $EF_{aer,super}=19-750$ ;  $EF_{aer,sub}=127-5100$ ;  $EF_{fog}=5-2400$ ) or the MIZ ( $EF_{aer,super}=60-2310$ ;  $EF_{aer,sub}=390-16000$ ;  $EF_{fog}=17-7400$ ) were assumed as the only (theoretical) source of CCHO and Na+, while higher values were found with the ice-free ocean ( $EF_{aer,super}=175-6800$ ;  $EF_{aer,sub}=1100-46000$ ;  $EF_{fog}=50-22000$ ) or open leads/polynyas ( $EF_{aer,super}=360-14000$ ;  $EF_{aer,sub}=2360-95000$ ;  $EF_{fog}=103-44600$ ).' (Lines 660-664)

Fig SI 4 and 5: First, the panel for pCCHO and a\_ph440 is not shown in Fig SI 4. Also, given some of these results seem like an important finding to the manuscript, the panels that are explicitly discussed in detail in the text should be presented as a main figure. The remaining panels could remain in the SI.

**Authors:**

**-missing panel**

We respectfully believe there is a misunderstanding on the reviewer's part. The panel for pCCHO and  $a_{ph}$ 440 is indeed presented, in the revised version now referred to as Figure 9a.

**-presentation as main figures**

We have shifted 'Figure SI 4' and 'Figure SI 5' from the supplementary information to the main manuscript, where they are now referred to as 'Figure 9' and 'Figure 10'.

**Literature cited by Reviewer 1:**

Alpert, P. A., Kilthau, W. P., O'Brien, R. E., Moffet, R. C., Gilles, M. K., Wang, B., Laskin, A., Aller, J. Y., and Knopf, D. A.: Ice-nucleating agents in sea spray aerosol identified and quantified with a holistic multimodal freezing model, Science Advances, 8, eabq6842, https://doi.org/10.1126/sciadv.abq6842, 2022.

Chen, Q., Mirrielees, J. A., Thanekar, S., Loeb, N. A., Kirpes, R. M., Upchurch, L. M., Barget, A. J., Lata, N. N., Raso, A. R. W., McNamara, S. M., China, S., Quinn, P. K., Ault, A. P., Kennedy, A., Shepson, P. B., Fuentes, J. D., and Pratt, K. A.: Atmospheric particle abundance and sea salt aerosol observations in the springtime Arctic: a focus on blowing snow and leads, Atmos. Chem. Phys., 22, 15263–15285, https://doi.org/10.5194/acp-22-15263-2022, 2022.

Huang, J. and Jaeglé, L.: Wintertime enhancements of sea salt aerosol in polar regions consistent with a sea ice source from blowing snow, Atmos. Chem. Phys., 17, 3699–3712, https://doi.org/10.5194/acp-17-3699-2017, 2017.

Kirpes, R. M., Bondy, A. L., Bonanno, D., Moffet, R. C., Wang, B., Laskin, A., Ault, A. P., and Pratt, K. A.: Secondary sulfate is internally mixed with sea spray aerosol and organic aerosol in the winter Arctic, Atmos. Chem. Phys., 18, 3937–3949, https://doi.org/10.5194/acp-18-3937-2018, 2018.

Kirpes, R. M., Bonanno, D., May, N. W., Fraund, M., Barget, A. J., Moffet, R. C., Ault, A. P., and Pratt, K. A.: Wintertime Arctic Sea Spray Aerosol Composition Controlled by Sea Ice Lead Microbiology, ACS Cent. Sci., 5, 1760–1767, https://doi.org/10.1021/acscentsci.9b00541, 2019.

May, N. W., Quinn, P. K., McNamara, S. M., and Pratt, K. A.: Multiyear study of the dependence of sea salt aerosol on wind speed and sea ice conditions in the coastal Arctic: ARCTIC SEA SALT AEROSOL, J. Geophys. Res. Atmos., 121, 9208–9219, https://doi.org/10.1002/2016JD025273, 2016.

**Reviewer: 2**

The authors collected samples from various aquatic and atmospheric sources during the PS106 campaign in May-June 2017. In this study they analyze and compare the combined carbohydrate concentrations, both dissolved and particulate, and discuss mechanisms for their enrichment in the SML and transfer into the air. Their findings add to a severely limited set of data on SML and CCHO concentrations in the Arctic region and have implications for both bio-chemical properties at the air-sea interface as well as CCN and INP formation in the air. The manuscript is well written, although some grammatical suggestions have been given. General and specific comments are given below which should be addressed prior to publication.

Authors: We thank the reviewer for the constructive suggestions and carefully revised the manuscript accordingly.

**General Comments:**

As I was able to see reviewer 1's comments before giving my review, I have tried to refrain from giving any duplicate suggestions, however please be aware that I am in complete agreement with all of their comments.

It would be nice if you could include information about the trajectories and age of the water masses you sampled from. Similar to air masses, arctic water in ice-free zones, MIZ and leads have long residence times of OM in the surface waters and thus will be impacted by the source of these waters. Did anyone take oxygen samples that you could include?

**Authors:**

**-trajectories and age of water masses**

We appreciate the reviewer's suggestion to expand the discussion on seawater history. Indeed, the trajectories and age of the water masses are important aspects that can potentially contribute to better understand the seawater concentrations. However, as this paper primarily addresses the diverse emission sources, sea-air-transfer, and atmospheric aging of marine carbohydrates, we believe that such considerations are beyond the scope of this study.

**-oxygen samples**

We did not conduct explicit oxygen measurements in our samples.

You compare this study to your other study in the southern ocean a lot. I would suggest to include a broader range of studies for comparison or include a good argument as to why you are comparing these two studies specifically.

Authors: We agree that the here performed study is compared a lot with our recent study (Zeppenfeld et al. 2021). The reason is that both studies were almost identical in terms of the experimental design to thoroughly compare the Arctic and the Antarctic with regards to the features discussed here. Both studies were conducted under similar meteorological conditions in polar regions, and our Southern Ocean study had the marked difference of lacking sea ice presence. This distinction simplified result interpretation in comparison to this Arctic study.

We agree that a comparison with other studies is beneficial, however we believe that other experimental design or analytical methodology would not allow to investigate the distinct behavior of polysaccharide emission and aging in the remote marine field. Also, we accounted similar approaches and their findings in the introduction and in the discussion, such as Hasenecz et al. (2019); Hasenecz et al. (2020); Burrows et al. (2016); Leck et al. (2013); etc.

We have incorporated this aspect to the last section of the introduction. The added sentence now reads: 'The complex nature of these primary emission mechanisms and subsequent atmospheric aging of marine CCHO in the Arctic Ocean are discussed in relation to our findings. Our Arctic results are collated with those from the Southern Ocean at the Antarctic peninsula during the austral summer, as presented in Zeppenfeld et al. (2021a) following a similar experimental design. While both polar locations are remote marine regions with comparable meteorological conditions during the sampling periods, the presence of Arctic sea ice adds another dimension of complexity to data interpretation.' (Lines 141-147)

During the ice floe camp, how far was your sampling location from RV Polarstern? As you are comparing aerosol measurements sampled on the ship with water samples taken nearby this is important, it would be good to describe the ice flow camp situation in general. How and why was the location or time chosen etc. Additionally, how far from the ice edge did you sample in the leads, this is important to know as later on you discuss the effects of meltwater on SML samples.

**Authors:**

**-distance sampling location on ice floe from RV Polarstern**

The sampling location typically ranged from 100-500 m from the RV Polarstern. This distance was chosen as a compromise to ensure, on one hand, that the sea spray aerosol particles sampled on the RV Polarstern related with the collected seawater, and on the other hand, to minimize anthropogenic impact from the ship, such as fuel exhausts and wastewater. We added this information to the manuscript: 'To minimize contamination from exhausts and wastewater, water samples were taken at distances greater than 100 m

from the ship. Seawater was collected either using a rubber boat or directly from the ice edge.' (Lines 159-161)

**-ice floe camp situation**

The ice floe camp constituted a two-weeks segment of the two-month-long PS106 campaign. Details of this period are available in Macke and Flores (2018) and Wendisch et al. (2018), as cited in our manuscript. Given that the camp's infrastructure did not influence our sampling or experimental approach, except for increased measures against anthropogenic contaminations, we deem this aspect not to be of primary relevance to our manuscript.

**-how and why was the location or time chosen**

The PS106 campaign covered late spring/early summer to gain experience for scientific operations on a drifting ice floe during this part of the year. Criteria for the choice may have included reachability for the AWI aircrafts POLAR 5 and POLAR 6 from Svalbard for collocated airborne measurements, as well as the floe's size, shape, safety considerations, and calculated drift trajectories.

**-distance to ice edge on the floe and relevance of meltwater on SML**

The reviewer's question likely pertains to samples collected in open leads or the MIZ. For other sea-icerelated sea surface compartments, proximity to the ice edge may not be significant. We generally aimed to maximize our distance from the ice edge, sampling from a rubber boat located as far away as feasible. Only occasionally did we sample directly at the ice edge, a method limited due to heightened safety precautions and additional personnel requirements. Given the rather cold temperatures within the pack ice region during this early part of the summer and the high, ocean-like salinities in the SML and bulk samples, we posit that open lead samples were only minimally influenced by meltwater during PS106 campaign. Conversely, MIZ samples were notably impacted by meltwater.

Figure SI 1. The zoomed in ice flow image is hard to orientate withing the larger cruise track image, please indicate where in the larger map the ice flow camp occurred.

Authors: We revised Figure SI1 and indicated where the ice flow occurred in the larger map by adding a dashed box and an arrow (see below).

**Specific Comments:**

Line 47: (primary marine aerosol particles)

Authors: Following the recommendation by reviewer 1, we replaced 'primary marine aerosol particles' with 'SSA particles'.

**Line 55: "appear" not "appears"**

Authors: Changed.

**Line 72: "suggested", "found" or similar, not assumed, assume suggests lack of evidence**

Authors: Thank you for pointing out this important difference in meaning. We replaced 'assumed' with 'suggested'.

**Line 83: give reference for bubble scavenging as this is different from bubble bursting**

Authors: We added Burrows et al. (2014) as a suitable reference. Furthermore, we rephrased slightly the sentence that now reads: 'Entrained air bubbles rise within the upper part of the water column collecting surface-active organics on the bubble surfaces from the bulk seawater (Burrows et al., 2014).' (Lines 87-89)

**Line 84: "within the bulk water and the SML"**

Authors: Changed.

Line 115: open leads can be much larger than several meters, would be good to define the difference between open leads and polynyas as these are often incorrectly used interchangeably, here it seems you use both because you are not sure which your study areas would be classified as, which is fine but should be addressed.

Authors: We agree with the reviewer. We extended the definitions of both the open leads and polynyas within the introduction of the manuscript: 'These encompass the open leads - sea ice fractures with variable widths ranging from several to hundreds of meters - and polynyas, which are larger, more persistent areas of open water within the pack ice.' (Lines 121-123)

Additionally, in section '2.1 Study area and field sampling', we have included a note about the absence of differentiation between open leads and polynyas in this study. The changed text now reads: 'Marine SML and corresponding bulk water samples were collected from various locations as shown in **Figure SI 1**. These include the ice-free ocean (four sampling events), open water areas within the pack ice (20 sampling events, without distinguishing between open leads and polynyas), the MIZ (five sampling events), and young and aged melt ponds (six sampling events).' (Lines 154-157)

**Line 117: "defined by a .."**

**Authors: Changed.**

Line 134: "eventually" and "disclose" not the best word choice, try "The complex nature of these primary emission mechanisms and subsequent atmospheric aging of marine CCHO in the Arctic Ocean are discussed in relation to our findings."

**Authors: Changed.**

**Line 144: how were ages of melt pond determined? How young is young?**

Authors: We categorized melt ponds as either 'young' or 'aged' based on visual characteristics, emphasizing that this designation is relative and not indicative of a specific age in numerical terms.

Samples from young melt ponds, which were smaller, bluish, and clear, were collected in mid-June. In contrast, aged melt pond samples, taken at the end of June or July, were larger, exhibited a darker blue to greenish hue, and appeared more turbid due to visible particulates and microalgae. While our two-month participation in the PS106 campaign didn't encompass the entire melt pond evolution until late summer, the rapid shift in their biological and chemical properties was evident. We've detailed our categorization criteria for young and aged melt ponds in the experimental section of our manuscript: 'Using visual characteristics, melt ponds were categorized as young (small, bluish, clear) or aged (larger, darker blue to greenish, and turbid with particulates and microalgae).' (Lines 157-159)

Line 157: what was the ambient air temperature when sampling with glass plate? It's good as a reader to know, as when air temperatures get low enough the SML samples can freeze on the glass plate before they are wiped off and may impact the carbohydrate measurements. However, it's likely that temperatures weren't this cold during the time of year you were there.

Authors: Due to its high salinity, seawater generally begins to freeze at temperatures below -1.8°C. The PS106 campaign occurred during the summer months, from May to July, with air temperatures hovering around the melting point. While there was no direct air temperatures measurements at the SML sampling site, continuous monitoring of air temperature took place on RV *Polarstern* at 29 m above sea level (Schmithüsen, 2018, 2019). During PS106, the median and minimum recorded air temperatures were -0.5°C and -7.6°C, respectively. As such, at no point during the PS106 campaign did temperatures drop low enough for the SML samples to immediately freeze upon contact with the glass plate. For adding clarity, we've incorporated the following sentence to the manuscript: 'Despite air temperatures during PS106 (median: -0.5°C; minimum: -7.6°C) hovering around or slightly below the freezing point of seawater, the SML remained unfrozen on the glass plate during sampling.' (Lines 168-170)

**Line 267: you use sea-ice (hyphenated) here but not elsewhere in the manuscript, be careful to homogenize your grammar choice**

Authors: We have consulted a native English speaker on the proper use of hyphens and adjusted the manuscript accordingly. While we use 'sea ice' without a hyphen, 'sea-ice-related' now includes two hyphens. We have ensured consistency throughout the manuscript.

Line 286: "eventually" used again. Just leave it out "The influence of the ...."

**Authors: Changed.**

Line 292: "Among all aqueous samples, regardless ... dCCHO (...) and pCCHO (...) concentrations were highly variable"

**Authors: Changed.**

Line 294: "However, the minimum, maximum and mean values of both dCCHO and pCCHO ranged within the same orders of magnitude"

Authors: Changed.

Line 307: The region comparisons you use for SML CCHO concentrations seem random, you should either compare with other arctic region studies, there have been quite a few studies of SML in the Fram Straight, or refer to overview studies which compare from a large variety of regions (e.g. Wurl et al. 2011.)

Authors: We acknowledge the reviewer's concern. To ensure the most comparable results for SML, we preferred the exclusive citation of literature using similar analytical methods, specifically chromatography. This criterion significantly narrowed the available options. To enhance the comparison on Arctic studies, we still added the findings by Gao et al. (2012). The revised text now reads: 'The lower SML concentrations from this study for the Arctic ice-free open ocean and lead/polynya samples align closely with several other investigations. Specifically, our results are comparable to Gao et al. (2012), who studied the SML of Arctic leads (meandCCHO, SML,Arctic leads = 163±104 µg L-1; meanpCCHO, SML, Arctic leads = 35±25 µg L-1; n=4), and Zeppenfeld et al. (2021), focusing on the ice-free part of the Southern Ocean west of the Antarctic peninsula during the austral summer (meandCCHO, SML, Southern Ocean = 48±63 µg L-1; meanpCCHO, SML, Southern Ocean = 72±53 µg L-1; n=18). Similarly, our data mirror findings from the tropical Cape Verde  $(mean_{dCCHO, SML, Cape Verde} = 85\pm 30 \ \mu g \ L^{-1};$  van Pinxteren et al., 2023) and the Peruvian upwelling region (meandCCHO, SML, Peru ≈ 92±32 µg L-1; Zäncker et al., 2017). Consequently, the Arctic MIZ and melt ponds, especially the aged ones with advanced microbiological activities, stood out with elevated CCHO within the Arctic and also compared to tropical and other polar regions.' (Lines 330-341)

Line 317: "80% of the SML samples were moderately or highly enriched in marine carbohydrates, with only a few cases of depletion (7 for dCCHO and 8 ..."

Authors: Changed.

**Line 319: I'm not sure stating the median concentration is very useful here**

Authors: We agree that stating the median has some disadvantage, however, given the broad range of values within each of the four sea-ice-related sea surface compartments, we believe the median offers the most accurate representation for comparing these groups. Thus, we have chosen to maintain the median values in this section.

Line 358: If release of melt water results in increases of CCHO then shouldn't you have found significantly higher mean values in your melt ponds and open leads compared to MIZ and ice-free zones? Galgani et al. 2016. in fact have done a similar study and did find this. It would be good for you to address that study and the possible relations and difference it has to your study.

**Authors:**

-expectation of higher mean values in melt ponds and open leads compared to MIZ and ice-free zones.

Of the four sea-ice-related sea surface compartments discussed in this manuscript, the MIZ and melt ponds have the highest meltwater input from sea ice. While melt ponds might eventually connect to the ocean below, contributing CCHO from melt ponds to open leads, this is likely at a smaller scale than in the MIZ. Therefore, we disagree with the reviewer's suggestion that open leads would have higher CCHO concentrations than the MIZ due to meltwater increasing the CCHO concentration.

**-comparison with Galgani et al. (2016)**

We concur with the reviewer on the significance of Galgani et al. (2016) in understanding the role of melt ponds and melting sea ice as sources of various dissolved organic components, such as dissolved organic carbon (DOC), dissolved uronic acids (DURA, a subgroup of DCCHO), dissolved hydrolysable amino acids (DHAA), transparent exopolymer particles (TEP) and Coomassie stainable particles (CSP), within the Arctic environment. However, their study primarily centered on freshwater melt ponds, open melt ponds, and open leads (termed 'open sea'), limiting comparisons to these two sea-ice-related sea surface compartments. Moreover, their research was carried out in the end of the summer during peak melting, while ours spanned May to July with still less advanced melting.

A pivotal observation from Galgani et al. (2016) highlighted the more labile nature of dissolved organic matter (DOM) in melt ponds compared to the older, more refractory DOM found in open leads, evidenced by the higher DURA/DOC and DHAA/DOC ratios. This insight aligns with our observation that melting sea ice and melt ponds serve as a productive source for the release of DCCHO within the pack ice zone. We highlighted this finding in the revised manuscript. The changed text now reads: 'Specific to the Arctic, the release of meltwater from the sea ice could be an additional source for carbohydrates in the SML, considering the production of CCHO, exopolymeric substances (EPS) and TEP by sea ice algae and bacteria as a protection strategy against freezing damage and fluctuating salinity in sea ice (Aslam et al., 2016; Krembs et al., 2002; Krembs and Deming, 2008). This aligns well with the finding by Galgani et al. (2016) who observed labile, fresh OM in the SML of melt ponds compared to the rather old, refractory nature of the SML in the surrounding open leads.' (Lines 385-390)

Line 366: Did you compare the salinity of the SML with the dCCHO and pCCHO concentrations? Increased biogenic material in the SML may result in a freshening of the SML. Additionally, Mari et al. 2012 found that in estuaries where there are large salinity gradients, the aggregation properties of TEP changed with salinity, a similar relationship between salinity and the phisico-chemical reactivity of TEP or CCHO may exist in arctic melt ponds and leads, and could show in the ratio of dissolved to particulate CCHO. This would additionally impact the ability for these particles to transfer into the air.

Authors: The aspect that the reviewer highlights is indeed compelling. We examined potential correlations between salinity and concentrations of dCCHO, pCCHO, individual monosaccharide constituents, and dCCHO/pCCHO ratios. This ambiguity could arise from the multifaceted nature of our samples, influenced by more than just salinity. Other factors, such as the history of water masses, nutrient availability, and microbial communities, weren't exhaustively explored as they were beyond this study's scope. We genuinely appreciate the reviewer's insight and believe a more focused approach with more samples from a tighter sampling area could be illuminating in future studies.

**Line 390: "The high Arctic..."**

Authors: Changed.

**Line 866: remove "in a nut shell", too informal, no need for it.**

Authors: Done.

**Literature cited by Reviewer 2:**

Galgani, L., Piontek, J. and Engel, A., 2016. Biopolymers form a gelatinous microlayer at the air-sea interface when Arctic sea ice melts. Scientific Reports, 6(1), p.29465.

Mari, X., Torréton, J.P., Trinh, C.B.T., Bouvier, T., Van Thuoc, C., Lefebvre, J.P. and Ouillon, S., 2012. Aggregation dynamics along a salinity gradient in the Bach Dang estuary, North Vietnam. Estuarine, Coastal and Shelf Science, 96, pp.151-158.

Wurl, O., Miller, L. and Vagle, S., 2011. Production and fate of transparent exopolymer particles in the ocean. Journal of Geophysical Research: Oceans, 116(C7).

Wurl, O., Wurl, E., Miller, L., Johnson, K. and Vagle, S., 2011. Formation and global distribution of sea-surface microlayers. Biogeosciences, 8(1), pp.121-135.

**References cited by the authors:**

Burrows, S. M., Ogunro, O., Frossard, A., Russell, L. M., Rasch, P. J., and Elliott, S.: A Physically Based Framework for Modelling the Organic Fractionation of Sea Spray Aerosol from Bubble Film Langmuir Equilibria, Atmospheric Chemistry and Physics, 14(24):13601–13629, https://doi.org/10.5194/acp-14-13601-2014, 2014.

Burrows, S. M., Gobrogge, E., Fu, L., Link, K., Elliott, S. M., Wang, H., and Walker, R.: OCEANFILMS-2: Representing coadsorption of saccharides in marine films and potential impacts on modeled marine aerosol chemistry, Geophysical Research Letters, 43, 8306–8313, https://doi.org/10.1002/2016GL069070, 2016.

Cunliffe, M. and Wurl, O.: Guide to best practices to study the ocean's surface., Marine Biological Association of the United Kingdom for SCOR, 2014.

Gao, Q., Leck, C., Rauschenberg, C., and Matrai, P. A.: On the chemical dynamics of extracellular polysaccharides in the high Arctic surface microlayer, Ocean Science, 8, 401–418, 2012.

Gong, X., Zhang, J., Croft, B., Yang, X., Frey, M. M., Bergner, N., Chang, R. Y.-W., Creamean, J. M., Kuang, C., Martin, R. V., Ranjithkumar, A., Sedlacek, A. J., Uin, J., Willmes, S., Zawadowicz, M. A., Pierce, J. R., Shupe, M. D., Schmale, J., and Wang, J.: Arctic warming by abundant fine sea salt aerosols from blowing snow, Nat. Geosci., 16, 768–774, https://doi.org/10.1038/s41561-023-01254-8, 2023.

Hasenecz, E., Jayarathne, T., Pendergraft, M. A., Santander, M. V., Mayer, K. J., Sauer, J., Lee, C., Gibson, W. S., Kruse, S. M., Malfatti, F., Prather, K. A., and Stone, E. A.: Marine bacteria affect saccharide enrichment in sea spray aerosol during a phytoplankton bloom, ACS Earth Space Chem., 4, 1638–1649, https://doi.org/10.1021/acsearthspacechem.0c00167, 2020.

Hasenecz, E. S., Kaluarachchi, C. P., Lee, H. D., Tivanski, A. V., and Stone, E. A.: Saccharide Transfer to Sea Spray Aerosol Enhanced by Surface Activity, Calcium, and Protein Interactions, ACS Earth Space Chem., 3, 2539–2548, https://doi.org/10.1021/acsearthspacechem.9b00197, 2019.

Herrmann, H., Schaefer, T., Tilgner, A., Styler, S. A., Weller, C., Teich, M., and Otto, T.: Tropospheric Aqueous-Phase Chemistry: Kinetics, Mechanisms, and Its Coupling to a Changing Gas Phase, Chem. Rev., 115, 4259–4334, https://doi.org/10.1021/cr500447k, 2015.

Kumai, M.: Arctic Fog Droplet Size Distribution and Its Effect on Light Attenuation, Journal of the Atmospheric Sciences, 30, 635–643, https://doi.org/10.1175/1520-0469(1973)030<0635:AFDSDA>2.0.CO;2, 1973.

Leck, C., Gao, Q., Mashayekhy Rad, F., and Nilsson, U.: Size-resolved atmospheric particulate polysaccharides in the high summer Arctic, Atmospheric Chemistry and Physics, 13, 12573–12588, https://doi.org/10.5194/acp-13-12573-2013, 2013.

Macke, A. and Flores, H.: The Expeditions PS106/1 and 2 of the Research Vessel POLARSTERN to the Arctic Ocean in 2017, Bremerhaven, Germany, 171 pp., https://doi.org/10.2312/BzPM\_0719\_2018, 2018.

van Pinxteren, M., Barthel, S., Fomba, K. W., Müller, K., Von Tümpling, W., and Herrmann, H.: The influence of environmental drivers on the enrichment of organic carbon in the sea surface microlayer and in submicron aerosol particles – measurements from the Atlantic Ocean, Elem Sci Anth, 5, https://doi.org/10.1525/elementa.225, 2017.

Schmithüsen, H.: Continuous meteorological surface measurement during POLARSTERN cruise PS106/1 (ARK-XXXI/1.1), https://doi.org/10.1594/PANGAEA.886302, 2018.

Schmithüsen, H.: Continuous meteorological surface measurement during POLARSTERN cruise PS106/2 (ARK-XXXI/1.2), https://doi.org/10.1594/PANGAEA.901179, 2019.

Verdugo, P., Alldredge, A. L., Azam, F., Kirchman, D. L., Passow, U., and Santschi, P. H.: The oceanic gel phase: a bridge in the DOM–POM continuum, Marine Chemistry, 92, 67–85, https://doi.org/10.1016/j.marchem.2004.06.017, 2004.

Wendisch, M., Macke, A., Ehrlich, A., Lüpkes, C., Mech, M., Chechin, D., Dethloff, K., Barientos, C., Bozem, H., Brückner, M., Clemen, H.-C., Crewell, S., Donth, T., Dupuy, R., Ebell, K., Egerer, U., Engelmann, R., Engler, C., Eppers, O., Gehrmann, M., Gong, X., Gottschalk, M., Gourbeyre, C., Griesche, H., Hartmann, J., Hartmann, M., Heinold, B., Herber, A., Herrmann, H., Heygster, G., Hoor, P., Jafariserajehlou, S., Jäkel, E., Järvinen, E., Jourdan, O., Kästner, U., Kecorius, S., Knudsen, E. M., Köllner, F., Kretzschmar, J., Lelli, L., Leroy, D., Maturilli, M., Mei, L., Mertes, S., Mioche, G., Neuber, R., Nicolaus, M., Nomokonova, T., Notholt, J., Palm, M., van Pinxteren, M., Quaas, J., Richter, P., Ruiz-Donoso, E., Schäfer, M., Schmieder, K., Schnaiter, M., Schneider, J., Schwarzenböck, A., Seifert, P., Shupe, M. D., Siebert, H., Spreen, G., Stapf, J., Stratmann, F., Vogl, T., Welti, A., Wex, H., Wiedensohler, A., Zanatta, M., and Zeppenfeld, S.: The Arctic Cloud Puzzle: Using ACLOUD/PASCAL Multi-Platform Observations to Unravel the Role of Clouds and Aerosol Particles in Arctic Amplification, Bull. Amer. Meteor. Soc., https://doi.org/10.1175/BAMS-D-18-0072.1, 2018.

---

## Author Response (AR2)

**Answer to the Editor:**

**Marine Carbohydrates in Arctic Aerosol Particles and Fog – Diversity of Oceanic Sources and Atmospheric Transformations**

Sebastian Zeppenfeld[1], Manuela van Pinxteren[1], Markus Hartmann[2], Moritz Zeising[3], Astrid Bracher[3,4], and Hartmut Herrmann[1]

1 Atmospheric Chemistry Department (ACD), Leibniz-Institute for Tropospheric Research (TROPOS), Leipzig, Germany
2 Atmospheric Microphysics (AMP), Leibniz-Institute for Tropospheric Research (TROPOS), Leipzig, Germany
3 Alfred-Wegener-Institute Helmholtz Centre for Polar and Marine Research, Bremerhaven, Germany
4 Institute of Environmental Physics, University of Bremen, Bremen, Germany

*Correspondence to*: Hartmut Herrmann (herrmann@tropos.de)

MS No.: egusphere-2023-1607
MS type: Research article

Dear Alex Huffman,

We thank you for the positive decision. We are hereby submitting a revised manuscript addressing your two minor comments.

**Editor: * Reviewer 1 asked whether there was an observed correlation between melt pond depth and measured CCHO concentration. You replied (paraphrasing) that there was some anecdotal differences, but that the number of measurements taken would fall below any statistical relevance. I agree that pointing out a formal correlation would be improper here, but it seems to me that there could be some value in reporting the qualitative observation you made and similar to what you summarized for the referee.**

Authors: We added this observation to the manuscript. The added sentence now reads: 'CCHO concentrations exhibited significant variability among the melt ponds, with higher concentrations observed in aged ponds (depths ranging from 40 cm to open-bottomed) compared to younger ones, where depths varied between 20 and 40 cm.' (Lines 325-328)

**Editor: * The abstract is very well written and nicely follows the format of the recently adopted ACP guidelines for abstract content (https://www.atmospheric-chemistry-and-physics.net/policies/guidelines_for_authors.html). After the new text you added to start the abstract, however, the length is now at ~335 words, which is beyond the new abstract limit of 250 words. I think with a little creativity and work you should be able to reduce wording somewhat to get to the abstract limit.**

Authors: We strongly shortened the abstract to comply with ACP's word limit. The revised abstract now contains 248 words and reads: 'Carbohydrates, originating from marine microorganisms, enter the atmosphere as part of sea spray aerosol (SSA) and can influence fog and cloud microphysics as cloud condensation nuclei (CCN) or ice nucleating particles (INP). Particularly in the remote Arctic region,

significant knowledge gaps persist about the sources, the sea-to-air transfer mechanisms, atmospheric concentrations, and processing of this substantial organic group. In this ship-based field study conducted from May to July 2017 in the Fram Strait, Barents Sea, and central Arctic Ocean, we investigated the sea-to-air transfer of marine combined carbohydrates (CCHO) from concerted measurements of the bulk seawater, the sea surface microlayer (SML), aerosol particles and fog. Our results reveal a wide range of CCHO concentrations in seawater (22–1070 µg L$^{-1}$), with notable variations among different sea-ice-related sea surface compartments. Enrichment factors in the sea surface microlayer (SML) relative to bulk water exhibited variability in both dissolved (0.4–16) and particulate (0.4–49) phases, with the highest values in the marginal ice zone (MIZ) and aged melt ponds. In the atmosphere, CCHO was detected in super- and submicron aerosol particles (CCHO$_{aer,super}$: 0.07–2.1 ng m$^{-3}$; CCHO$_{aer,sub}$: 0.26–4.4 ng m$^{-3}$) and fog water (CCHO$_{fog,liquid}$: 18–22000 µg L$^{-1}$; CCHO$_{fog, atmos}$: 3–4300 ng m$^{-3}$). Enrichment factors for sea-air transfer varied based on assumed oceanic emission sources. Furthermore, we observed rapid atmospheric aging of CCHO, indicating both biological/enzymatic processes and abiotic degradation. This study highlights the diverse marine emission sources in the Arctic Ocean and the atmospheric processes shaping the chemical composition of aerosol particles and fog.' (Lines 18-36)